# Future changes in coastal upwelling and biological production in eastern boundary upwelling systems

Tianshi Du [1], Shengpeng Wang [1] ✉, Zhao Jing [1,2], Lixin Wu [1,2], Chao Zhang [3] & Bihan Zhang[4]

Upwelling along oceanic eastern boundaries has attracted significant attention due to its profound effects on ocean productivity and associated biological and socioeconomic implications. However, uncertainty persists regarding the evolution of coastal upwelling with climate change, particularly its impact on future biological production. Here, using a series of state-of-the-art climate models, we identify a significant seasonal advancement and prolonged duration of upwelling in major upwelling systems. Nevertheless, the upwelling intensity (total volume of upwelled water) exhibits complex changes in the future. In the North Pacific, the upwelling is expected to attenuate, albeit with a minor magnitude. Conversely, in other basins, coastal upwelling diminishes significantly in equatorward regions but displays a slight decline or even an enhancement at higher latitudes. The climate simulations also reveal a robust connection between changes in upwelling intensity and net primary production, highlighting the crucial impact of future coastal upwelling alterations on marine ecosystems.

The coastal upwelling along the eastern boundary of ocean basins, driven by equatorward winds parallel to the coastline, facilitates the vertical transport of nutrient-rich water to the surface[1,2]. This nutrient influx nourishes marine ecosystems, fostering highly productive and biodiverse "hotspots"[3,4], which are particularly pronounced in the four major eastern boundary upwelling systems (EBUSs): California (CalCS), Canary (CanCS), Humboldt (HCS), and Benguela (BCS) Current Systems (Fig. 1a, b). Significantly, these EBUSs cover less than 1% of the global ocean surface but provide up to 7% of the global marine primary production and 20% of the world's capture fisheries[5]. Hence, accurately predicting how coastal upwelling will respond to anthropogenic climate change and its subsequent impact on biological productivity is of utmost importance.

Conventionally, wind-induced Ekman transport has been regarded as the primary mechanism driving upwelling in the EBUSs[6].

Consequently, Bakun hypothesized that intensified land-sea thermal contrast under future warming conditions would produce stronger pressure gradients, thereby enhancing coastal upwelling in the EBUSs by fortifying alongshore winds[7]. However, recent studies have underscored the impact of poleward migration of the high-pressure systems on future changes in coastal upwelling, projecting an intensified upwelling in higher latitudes and weakened upwelling in lower latitudes[8]. Besides, under greenhouse warming, an earlier and prolonged upwelling season was also projected in the high latitudes of EBUSs, while a different trend was identified in the low latitudes[9]. These studies equate changes in upwelling with changes in the upwelling-favorable winds. However, a meta-analysis revealed substantial diversity in the responses of regional coastal upwelling to anthropogenic forcing, even under universally increasing wind trends in most EBUSs[10–16]. In particular, future changes in stratification[14,15,17]

[1]Laoshan Laboratory, Qingdao, China. [2]Frontier Science Center for Deep Ocean Multispheres and Earth System (FDOMES) and Key Laboratory of Physical Oceanography, Ocean University of China, Qingdao, China. [3]FDOMES and Key Laboratory of Marine Environment and Ecology, Ministry of Education, Ocean University of China, Qingdao, China. [4]College of Marine Life Sciences, Department of Marine Ecology, Ocean University of China, Qingdao, China. ✉ e-mail: spwang1@qnlm.ac

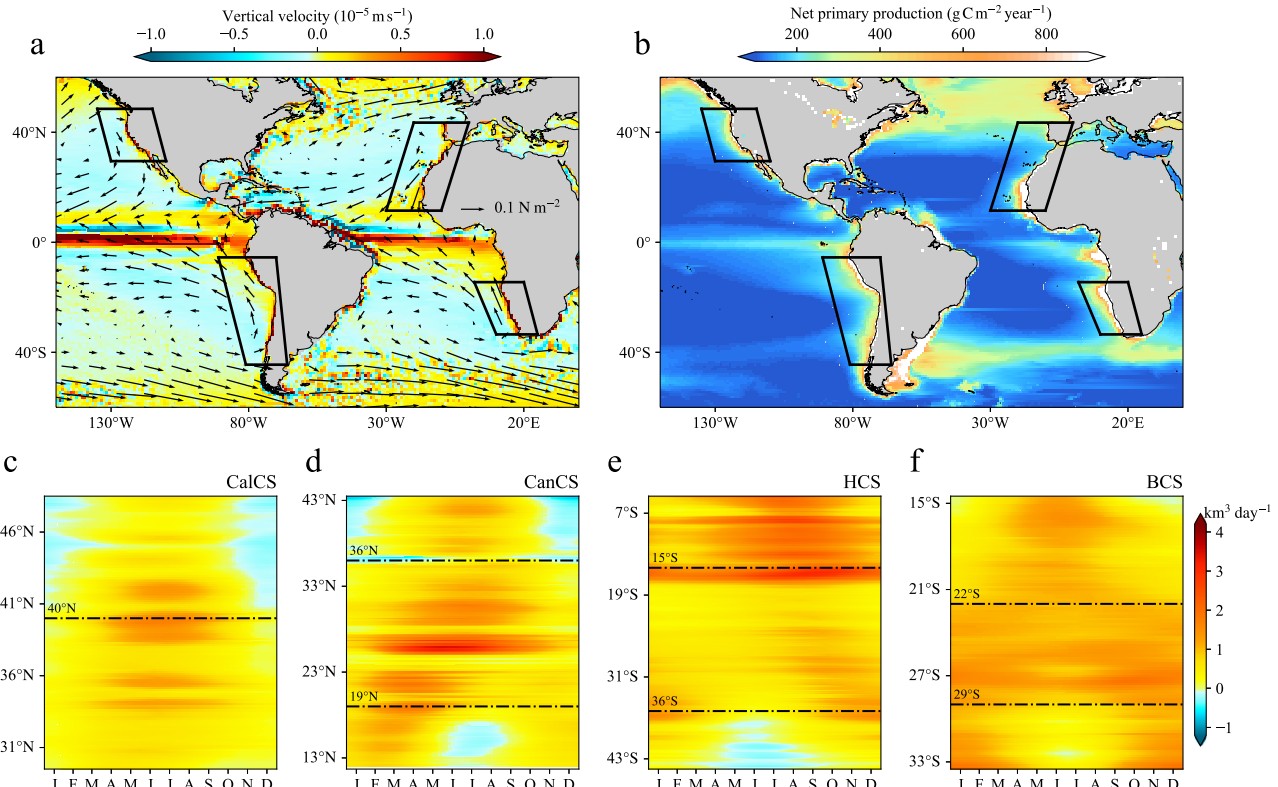

**Fig. 1 | Upwelling processes and primary productivity in the oceanic eastern boundaries. a** The vertical velocity at 50 m obtained from the high-resolution Community Earth System Model simulation (CESM-H) spanning from 2003 to 2022, with superimposed wind stress for the same period. **b** Observed vertically integrated primary production (see "Validation datasets" in Methods) during the same period from 2003 to 2022. **c–f** Hovmöller diagrams of the upwelling index (UI) in different ocean basin's upwelling systems, namely the California (CalCS), Canary (CanCS), Humboldt (HCS), and Benguela (BCS) Current Systems during 1992–2022 derived from CESM-H.

and geostrophic transport[16,18] could surpass the impact of winds, playing an important role in modulating the coastal upwelling. Therefore, considerable uncertainty persists regarding the evolution of coastal upwelling with climate change in terms of the intensity and timing[2,9].

In addition to the modifications in upwelling itself, our current understanding of the potential impacts of its changes on productivity in EBUSs remains limited. It is generally thought that greenhouse warming should reduce the productivity in the EBUSs by inhibiting the upward nutrient supply due to intensified stratification, which is supported by a majority of studies[14,15,19–21]. However, a lack of consensus on future productivity projections derived from climate models challenges the notion that stratification is the primary determinant in predicting forthcoming productivity changes in the EBUSs[1]. Evidence exists that the coastal upwelling contributes nonnegligible to the long-term changes of productivity in some EBUSs[22,23]. Besides, potential alterations in the nitrate content of upwelling source water could also exert a significant influence on productivity[23–25]. However, how the evolving dynamics of upwelling affect the nutrient supply and productivity in the major EBUSs remains uncertain.

The limited observational record length introduces considerable uncertainties when analyzing the impact of global warming on coastal upwelling using historical data[1]. Alternatively, state-of-the-art climate models, which have outputs of vertical velocity, offer an approach to directly evaluate the response of coastal upwelling to global warming. In this study, we employ an unprecedented high-resolution Community Earth System Model simulation (referred to as CESM-H) (see "Climate simulations" in Methods) combined with a series of current-generation climate simulations participated in the Coupled Model Intercomparison Project Phase 6 (CMIP6) (see "Climate simulations" in

Methods) to evaluate the future changes of coastal upwelling and its impact on the biological productivity in the major EBUSs under a high carbon emission scenario.

## Results

### Seasonal advancement and prolonged upwelling season

We begin with a broad view of the climatological coastal upwelling in the EBUSs. Consistent with the existing theoretical arguments and simulations[8,18,26,27], the coastal upwelling index (UI, see "Upwelling timing and intensity" in Methods) in the major EBUSs by CESM-H demonstrates significant seasonal variability. Yet, this seasonality varies within individual EBUSs depending on latitudes (Fig. 1c–f). In the poleward regions (CalCS-P, CanCS-P, HCS-P and BCS-P, see "Region selection" in Methods), upwelling displays a consistent and significant seasonal cycle, with its largest value occurring in summer (June-July-August for Northern Hemisphere and December-January-February for Southern Hemisphere) and smallest values in winter. Conversely, in the equatorward regions (CanCS-E, HCS-E, and BCS-E), the seasonal upwelling cycle exhibits contrasting patterns, with maximum values in winter and minimum values in summer. The central regions (CalCS-C, CanCS-C, HCS-C, and BCS-C), situated between the poleward and equatorward regions, exhibit strong permanent upwelling but weak seasonal variability across all EBUSs. In addition, the upwelling also displays distinct seasonality in terms of timing and duration: most regions demonstrate persistent annual upwelling, while three poleward regions (e.g. CalCS-P, CanCS-P, and HCS-P) exhibit positive upwelling from early spring to fall lasting for 250–270 days, and one equatorward region (e.g. CanCS-E) displays positive upwelling from early fall to the following spring lasting for nearly 300 days (Supplementary Fig. 1). The simulated climatology of coastal upwelling in

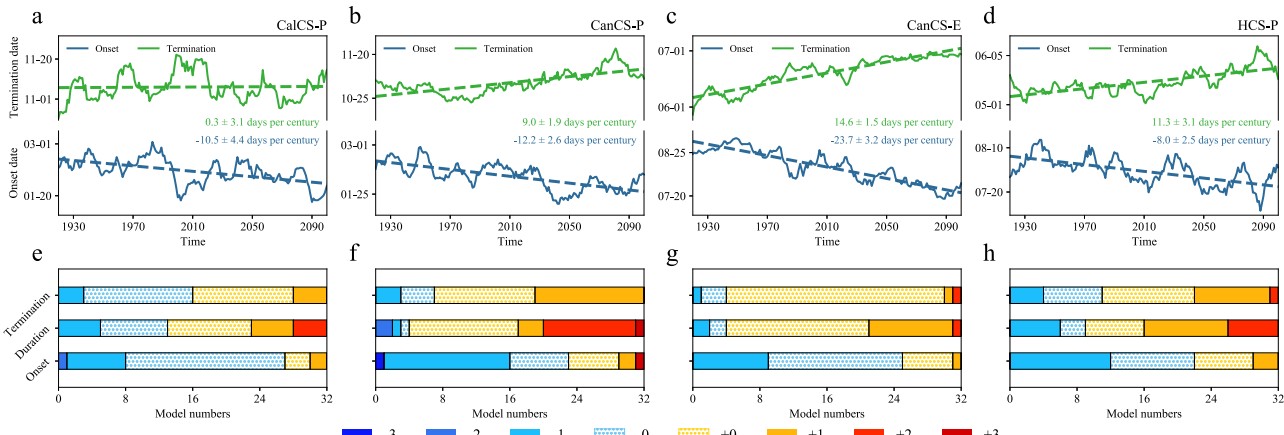

**Fig. 2 | Projected changes in upwelling timing of the eastern boundary upwelling systems (EBUSs) under a high carbon emission scenario. a** The time-series (solid) and linear trends (dotted) for the onset date (blue) and termination date (green) of upwelling season within the poleward region in California Current System (CalCS-P) derived from the daily outputs of the high-resolution Community Earth System Model simulation (CESM-H) during 1920–2100. The y-axis represents the onset and termination dates. The extent of the area between the two lines illustrates the duration of the upwelling season. **b–d** Same as **a** but for the onset date and termination date in the poleward region in Canary Current System (CanCS-P), equatorward region in CanCS (CanCS-E), and poleward region in Humboldt Current System (HCS-P). **e–h** The onset, termination, and duration differences derived from the climate simulations in the Coupled Model Inter-comparison Project Phase 6 (CMIP6) between 2071–2100 and 1920–1949. The positive values (+1, +2, and +3; warm color) indicate the months in which the onset and termination of the upwelling season have advanced, whereas the negative values (−1, −2, and −3; cool color) represent the months in which the termination of the upwelling season have delayed. The +0 and −0 (hatched) also indicate the advancement and delay of the upwelling onset and termination dates but with the shifting rate less than one month. As for the duration, the positive and negative values correspond to the months that the upwelling season has prolonged and shortened, respectively. The x-axis represents the model numbers.

CESM-H closely aligns with that observed in the ocean reanalysis (Supplementary Fig. 2, see "Validation datasets" in Methods), albeit exhibiting a slightly stronger magnitude, potentially attributed to the higher spatial resolution of CESM-H. This agreement instills confidence in the capacity of CESM-H to project the coastal upwelling changes under greenhouse warming.

To quantitatively assess future changes in the timing of the upwelling season, we evaluate the shifting rates of upwelling onset and termination date by utilizing the daily vertical velocity output from CESM-H (see "Upwelling timing and intensity" in Methods). Figure 2a–d shows the linear trend (see "Trend analysis" in Methods) of the onset and termination date of the upwelling season for the regions with a seasonal upwelling. There is an obvious seasonal advancement and extension of upwelling season in both Hemispheres. In particular, the earlier-shifting trend of upwelling onset is most evident in the Northern Hemisphere, with a rate of 10.5 ± 4.4 days per century in the CalCS-P, 12.2 ± 2.6 days per century in the CanCS-P, and 23.7 ± 3.2 days per century in the CanCS-E, respectively. As for the Southern Hemisphere, the upwelling onset features a more minor earlier shift in the HCS-P compared to the Northern Hemisphere counterparts, with a rate of 8.0 ± 2.5 days per century. Concurrently, a later shifting in upwelling termination is also identified in these regions, which is statistically significant in CanCS-P, CanCS-E, and HCS-P. Consequently, the combination of earlier onset and delayed termination leads to an extended duration of the upwelling season in the CanCS-P, CanCS-E, and HCS-P regions, with a rate of 8.2 ± 1.2% per century (21.2 ± 3.1 days per century), 13.6 ± 1.4% per century (38.3 ± 4.0 days per century) and 6.9 ± 1.6% per century (19.3 ± 4.5 days per century), respectively. The prolonged duration of upwelling in the CalCS-P is almost entirely attributed to the advancement of upwelling onset, resulting in a more modest trend of 4.2 ± 2.7% per century or 10.8 ± 6.9 days per century. Additionally, the regions with permanent upwelling are projected to maintain persistent annual upwelling throughout the 21st century, indicating no anticipated changes in the timing and duration of the upwelling season (Supplementary Fig. 3). Notably, the upwelling seasonal cycle exhibits cross-shore discrepancies (Supplementary Fig 4), which might be attributed its diverse dynamic mechanisms[23,28,29].

However, the secular changes in onset and termination dates demonstrate a similarity pattern, which is not sensitive to the width of the upwelling band selected, providing strong support for the validity of our conclusions.

To confirm whether the seasonal advance and extension of upwelling projected by CESM-H is model-dependent, we further assess the shifting rates of upwelling seasonal timing using CMIP6 models. Notably, quantifying seasonal timing changes was not feasible using the monthly output from the CMIP6 models, particularly for temporal shifts occurring within a timeframe shorter than one month. Nevertheless, we were able to assess these changes by examining variations in upwelling during the transitional month (see "Shifting definition in monthly data" in Methods). Using the CMIP6 dataset, the results consistently show a significant earlier-shifting and longer-extending trend for the upwelling season both in the Pacific and Atlantic seasonal upwelling regions (Fig. 2d–f). In the Pacific basin regions, the CalCS-P and HCS-P, a majority of models (27 out of 32 for CalCS-P; 22 out of 32 for HCS-P) project an earlier onset of upwelling, with a significant proportion (19 out of 32 for CalCS-P; 23 out of 32 for HCS-P) anticipating a prolonged duration of the upwelling season. A similar scenario is observed in the Atlantic basin, where more than 72% (23 out of 32) and 78% (25 out of 32) of models successfully captured the trend of earlier upwelling occurrence in the CanCS-P and CanCS-E, respectively. Moreover, over 87% (28 out of 32) of models simulated prolonged duration of upwelling season in both CanCS-P and CanCS-E. Therefore, the evident seasonal advancement and extended duration of upwelling are not specific to CESM-H but are qualitatively reproduced by CMIP6 models.

To elucidate the potential mechanism underlying the advancement of the upwelling season, we have decomposed the upwelling during the onset month into components associated with the wind-driven Ekman transport and geostrophic transport (see "Decomposition of upwelling" in Methods). The upwelling during the onset month across all seasonal upwelling regions exhibits a consistent and significant increasing trend, indicating an earlier occurrence of upwelling events (Supplementary Fig. 5). However, the primary contributor varies among different ocean basins. In the Pacific basin, the advancement

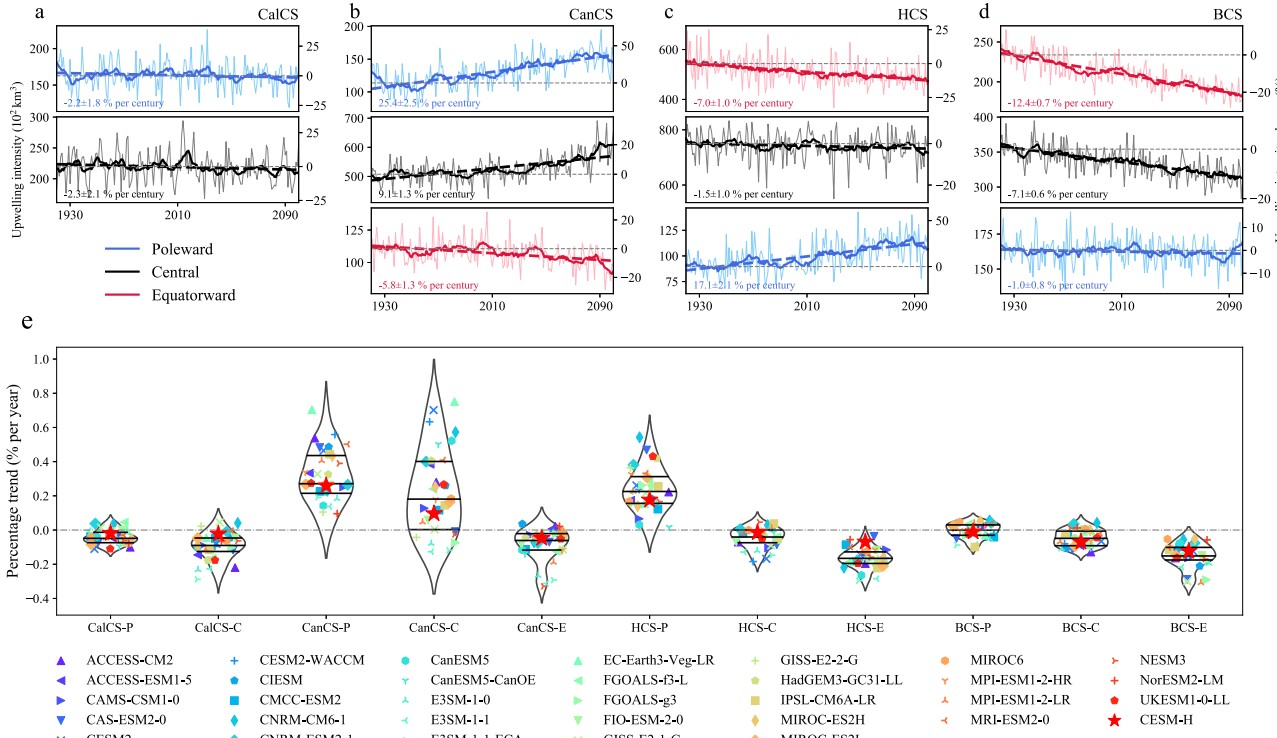

**Fig. 3 | Projected changes in upwelling intensity in eastern boundary upwelling systems (EBUSs). a** The secular changes of the upwelling intensity in the California Current Systems (CalCS) derived from daily high-resolution Community Earth System Model simulation (CESM-H). The thin colored line represents the inter-annual variability of upwelling intensity, while the thick colored line is smoothed by a 10-year 1-order Savitzky-Golay filter[55], representing the decadal changes. The dashed colored line denotes the linear trend during 1920–2100 with its slope and standard error shown in the text. The gray dashed line marks the climatological value (1920-1949), with the y-axis on the right-hand side of each panel showing the relative change concerning its climatological value. **b**–**d** Same as **a** but for results in

the Canary (CanCS), Humboldt (HCS), and Benguela (BCS) Current Systems. The red, black, and blue lines represent the secular changes of upwelling intensity in the equatorward (-E), central (-C), and poleward (-P) regions in individual EBUS, respectively. **e** The violin plot of the linear trends of the upwelling intensity in all subregions of EBUS derived from the climate simulations in the Coupled Model Intercomparison Project Phase 6 (CMIP6). The width of the plot at each value represents the density of data points, with wider sections indicating higher density. Within each plot, the central line represents the median value, flanked by the first and third quartiles. Trends from the monthly CESM-H (red star) and individual simulations are depicted as different symbols, with outliers removed from the plot.

of the upwelling season is primarily attributed to changes in Ekman transport, illustrating the dominant role of wind variations. Conversely, in the Atlantic basin, geostrophic transport surpasses Ekman transport and promotes the upwelling intensification, thereby advancing the upwelling season (Supplementary Fig. 5). The enlarged Ekman transport in the Pacific basin is governed by the intensified alongshore wind stress, resulting from heightened land-sea pressure gradients (Supplementary Fig. 6). However, the pressure gradient changes are not caused by intensified land-sea thermal contrast as suggested by Bakun[7], but are predominantly influenced by variations in atmospheric pressure within the ocean interior (Supplementary Fig. 6). This suggests that the intensified alongshore wind during the spring season in the Pacific basin may be attributed to large-scale atmospheric circulation changes, potentially originating from meridional shifts of the Intertropical Convergence Zone (ITCZ) in a warming climate[30].

## Complex changes in upwelling intensity

The changes in upwelling timing and duration across EBUSs do not extend to the future changes in upwelling intensity, which is defined as the total volume of upwelled water (Fig. 3, see "Upwelling timing and intensity" in Methods). The trends of upwelling intensity in major EBUSs exhibit a pronounced latitudinal-dependent pattern. Specifically, the CanCS-P, located in the poleward portion of the North Atlantic, is projected to experience an increase in upwelling intensity by 25.4 ± 2.5% per century. Similarly, the CanCS-C, situated at lower latitudes, also demonstrates an increasing trend in upwelling intensity, albeit with a comparatively weaker magnitude (9.1 ± 1.3% per century)

compared to the CanCS-P. In contrast, the CanCS-E, positioned in the equatorward portion, exhibits a significant negative trend (−5.8 ± 1.3% per century) in upwelling intensity. The HCS, located in the South Pacific, exhibits a similar pattern, characterized by a significant increase in upwelling intensity in its poleward region (17.1 ± 2.1% per century) and a decreasing trend of upwelling intensity in its equatorward region (−7.0 ± 1.0% per century). As for its central region (HCS-C), the upwelling intensity exhibits a weaker declining trend, with a magnitude of −1.5 ± 1.0% per century. Similar to the HCS, a decreasing trend in upwelling intensity is observed in both the equatorward and central regions of BCS, with a more pronounced magnitude in the former (−12.4 ± 0.7% per century) compared to the latter (−7.1 ± 0.6% per century). In contrast to the increasing trend observed in poleward regions in CanCS and HCS, BCS-P exhibits a slightly decreasing trend (−1.0 ± 0.8% per century) in upwelling intensity. This can be attributed to its lower-latitude location, associated with the geographical constraints imposed by the African continent. The CalCS differs from other EBUSs in that both poleward and central regions exhibit marginally declining upwelling intensity (−2.2 ± 1.8% per century for CalCS-P and −2.3 ± 2.1% per century for CalCS-C), which are statistically insignificant at the 95% confidence level. Besides, we remark that the future changes in upwelling intensity in most regions could be reproduced within the 50-km-wide coastal upwelling band, albeit with distinct magnitudes (Supplementary Fig. 7). However, large discrepancies are observed in certain EBUS, particular in the BCS-C. Specifically, the upwelling intensity in BCS-C exhibits a pronounced declining trend within the 200-km-wide band, while displaying a subtle decrease at a

distance of 50 km from the coastline. This indicates that the upwelling intensity might also exhibit a divergent distribution along the cross-shore direction in BCS-C. These divergent changes in upwelling intensity could be attributed to complex factors, including the divergent changes in along-shore wind, stratification, and eddy activities[12,23,28,29,31]. A comprehensive understanding of the spatial variability of the upwelling changes as well as their dynamic deserves further investigation in future studies.

To further test the robustness of the trends of upwelling intensity in the EBUSs projected by CESM-H, we compare the results from the CESM-H with those models from the CMIP6. Due to the substantial storage burden, only monthly mean vertical velocity is provided by CMIP6 models. Therefore, we approximate the upwelling intensity in both CMIP6 models and CESM-H by utilizing the monthly averaged vertical velocity. However, the impact of using monthly instead of daily data does not qualitatively affect future changes in upwelling intensity, as revealed by CESM-H (Supplementary Fig. 8), giving us confidence in its reliability for projecting future changes to upwelling intensity. The projected upwelling intensity in most CMIP6 models exhibits qualitative consistency with that of CESM-H across the major EBUSs (Fig. 3e). Particularly, over 80% (26 out of 32) of models project a reduction in upwelling intensity in the CalCS. In the North Atlantic and South Pacific, more than 75% (24 out of 32) and 78% (25 out of 32) of models successfully replicate the amplification of upwelling in their poleward region and the attenuation of upwelling in their equatorward region. Regarding the BCS in the South Atlantic, models also successfully reproduce the latitudinal pattern in the secular changes of upwelling intensity: all models indicate a significant decrease in its equatorward region, with 78% (25 out of 32) showing a moderate decreasing trend in its central region, while only 53% (17 out of 32) demonstrate a weak declining trend in its poleward region. Thus, the projected changes in upwelling intensity vary among different EBUSs but remain consistent across different models within individual EBUSs, providing compelling evidence for its robustness. However, there are also quantitative differences among the projected trends in CESM-H and individual CMIP6 simulations, potentially attributed to the heterogeneous capacity of climate models in accurately representing the upwelling dynamics.

The response of upwelling intensity to greenhouse warming in the EBUSs is primarily attributed to changes in vertical velocity at 50 m (Supplementary Fig. 9a). Furthermore, these changes in upwelling intensity are predominantly associated with modifications in geostrophic transport across most upwelling regions (Supplementary Fig. 10), aligning with recent studies[16]. Specifically, in certain EBUSs such as the HCS-C, BCS-C, and BCS-P, despite projected increases in Ekman transport in the future, the overall upwelling intensity shows a decreasing trend. This reversal is caused by the strong offsetting effect of geostrophic transport, underscoring its significant influence. However, as latitudes increase, the dominant influence of geostrophic transport gradually diminishes, yielding to the prominence of wind-induced processes. The intensified Ekman transport at the poleward regions might be attributed to the poleward migration of atmospheric systems due to greenhouse warming[8]. However, future changes in geostrophic transport are influenced by various factors, including asymmetrical ocean warming, sea ice melting, etc.[32,33]. Consequently, distinct patterns of geostrophic transport are anticipated to manifest in individual EBUSs, warranting further investigation in future studies.

## Discussion

Our study unveils a consistent seasonal advancement and prolonged coastal upwelling in response to greenhouse warming. This provides valuable insights overlooked by previous studies as they predominantly focus on the future annual-mean upwelling changes while neglecting the seasonal diversity in future coastal upwelling patterns[14,16,31,34]. In addition, significant disparities were found between the long-term changes in upwelling intensity and annual-mean

upwelling for the seasonal upwelling regions, with even a reversed long-term trend (Supplementary Fig. 9b). It thus suggests that the annual-mean upwelling is not representative of the total upwelled water, which may further lead to misunderstandings of the projection of the biological productivity.

The findings of our study hold direct implications for projecting the net primary productivity (NPP) within the EBUSs under a warming climate. Considering the augmented (diminished) intensity of upwelling will have consequential impacts on the upward transportation of nutrients, future fluctuations in upwelling intensity are expected to exert influences on primary production in the EBUSs[22,35]. To test this hypothesis, we quantify the long-term changes in the upward nutrient transport and NPP based on the Earth System Models (ESMs) in CMIP6. As expected, the long-term trend of vertical nutrient transport (VNT, see "Computation of VNT" in Methods) in a majority of upwelling regions is primarily attributed to the changing upwelling intensity (Fig. 4a). Moreover, the VNT trends derived from the ESMs exhibit a robust alignment with those of the upwelling intensity across various regions (Fig. 4b). In particular, in CanCS-P, CanCS-C and HCS-P, where the projected increase in upwelling intensity is significant, the t-test reveals that a majority of models (10 out of 14 for CanCS-P, 12 out of 14 for CanCS-C, and 12 out of 14 for HCS-P) project significant increase in VNT. Similarly, in the remaining Northern Hemisphere regions, CalCS-P, CalCS-C, and CanCS-E, where a decrease in upwelling intensity is expected, more than 71% (10 out of 14), 93% (13 out of 14) and 100% (14 out of 14) of models project a significant decrease in VNT, respectively. Additionally, in the Southern Hemisphere upwelling regions, a majority of models (12 out of 14) indicate a significant decline in the VNT within the equatorward and central regions, corresponding to the negative trend of upwelling intensity observed in these regions.

Moreover, the ESMs projecting an increasing (decreasing) trend of VNT generally exhibit an enlarged (suppressed) primary production (Fig. 4c, see "NPP during the upwelling season" in Methods). The inter-model correlation coefficient between the trends of VNT and NPP reaches up to 0.64, statistically significant at the 99% confidence levels, providing further evidence that changes in upwelling intensity have a corresponding impact on biological productivity in the EBUSs. This close relationship between future changes in VNT and NPP is also observed in each EBUS (Supplementary Fig. 11), further underscoring the crucial role of upwelling intensity in governing the response of primary productivity to greenhouse warming in the EBUSs.

It is generally believed that the intensified stratification in a warming climate could reduce primary production by reducing the upper-layer nutrient supply[14,19]. However, limited consensus was exhibited across EBUSs on the relationship between changes in stratification and productivity (Supplementary Fig. 12). This implies that upwelling changes may overwhelm stratification changes, thus governing the ecosystem's productivity. Notably, changes in stratification could also affect the intensity and nutrient transport of upwelling by altering their source depths, influencing future productivity and potentially leading to unknown ecological and socio-economical effects[15]. However, this complex interaction cannot be easily isolated and is not addressed in this study.

Finally, it is important to acknowledge the limitations of the CMIP6 models used in this study due to their coarse resolution. Such low-resolution models can introduce significant deviations when representing the coastal upwelling and further affect the accuracy of biogeochemical variables[26,31,36–38]. Additionally, the coarse resolution is unable to resolve the smaller-scale oceanic processes (e.g., eddy activities, coastal trapped waves), which can have a substantial impact on the primary productivity in the EBUSs[16,23,25,31,39–43]. Understanding the changes in these smaller-scale processes and their contributions to primary productivity under global warming is crucial. Therefore, there is a substantial scope for enhancing current models[44,45]. Furthermore,

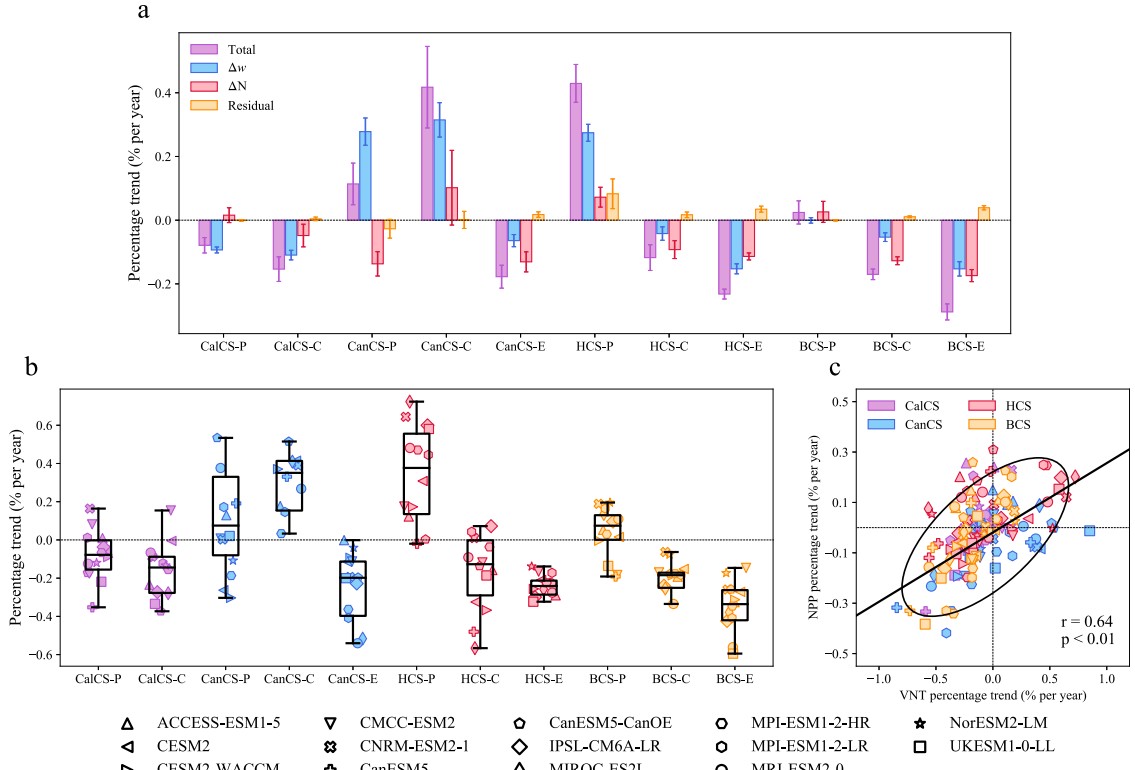

Symbol legend:

| Symbol | Model | Symbol | Model | Symbol | Model | Symbol | Model | Symbol | Model |
|---|---|---|---|---|---|---|---|---|---|
| △ | ACCESS-ESM1-5 | ▽ | CMCC-ESM2 | ⬠ | CanESM5-CanOE | ○ | MPI-ESM1-2-HR | ★ | NorESM2-LM |
| ◁ | CESM2 | ⊗ | CNRM-ESM2-1 | ◇ | IPSL-CM6A-LR | ○ | MPI-ESM1-2-LR | □ | UKESM1-0-LL |
| ▷ | CESM2-WACCM | ✚ | CanESM5 | ◇ | MIROC-ES2L | ○ | MRI-ESM2-0 | | |

**Fig. 4 | Ecological implications of coastal upwelling in eastern boundary upwelling systems (EBUSs) derived from the Earth System Models (ESMs) in the Coupled Model Intercomparison Project Phase 6 (CMIP6). a** Decomposition of vertical nutrient transport (VNT) changes (purple) into components associated with the upwelling effects ($\Delta w$, blue), nutrient effects ($\Delta N$, red), and interaction effects (yellow) in the poleward (-P), central (-C), and equatorward (-E) regions in California (CalCS), Canary (CanCS), Humboldt (HCS), and Benguela (BCS) Current Systems. The error bars denote the standard error across models. **b** Boxplots of linear trends of VNT in all subdomains of EBUSs during 1920-2100. The line in the middle of each box represents the median value, with the lower and upper bounds of each box indicating the first and third quartiles, respectively. The whiskers extend to 1.5 times the interquartile range. Individual ESMs are depicted as different symbols, with outliers removed from the plot. **c** Scatterplot of VNT changes and net primary production (NPP) changes in individual ESMs. Linear fit (solid black line) and 95% confidence ellipse are shown together with a correlation coefficient $r$ and corresponding $P$ value. K-means clustering algorithm[60] was first applied to remove the simulation anomalies.

it is noteworthy that the monthly outputs of the CMIP6 models employed in this study do not encompass certain high-frequency upwelling events, which might also exert a substantial impact on the biological production in the EBUSs[46,47]. Coupled with a reliable biogeochemical component, CESM-H could improve our knowledge of the response of EBUS ecosystems to greenhouse warming and its underlying dynamics, enabling targeted interventions to safeguard and sustain these vital marine ecosystems.

## Methods
### Climate simulations
The CESM-H simulation utilized in this study is based on version 1.3 of CESM. The model features a 0.25° resolution and 30 vertical levels for the atmosphere component and a 0.1° horizontal resolution and 62 vertical levels for the ocean component. For the oceanic component, the vertical grid space increases from 10 m near the surface to 250 m near the bottom. In particular, the thickness of each layer is identically set as 10 m in the upper 150 m, below which the thickness gradually increases with depth. The CESM-H includes a 500-year-long pre-industrial control (PI-CTRL) simulation and a 250-year-long historical and future transient simulation (HF-TNST). The HF-TNST simulation branches off from the 250th year of the PI-CTRL and incorporates historical forcings from 1850 to 2005 and representative concentration pathway 8.5 (RCP8.5) forcings from 2006 to 2100. For further details on the CESM-H simulation, please see the overview paper by Chang et al.[48]. Daily averaged outputs of vertical velocity at 50 meters are saved during 1920–2100.

Moreover, 32 climate models from CMIP6, which output vertical velocity, are employed in this study (Supplementary Table 1). The simulations incorporated historical forcings from 1850 to 2014 and utilized socioeconomic pathway 5-8.5 (SSP5-8.5, aligned with RCP8.5 adopted by CESM-H but accounting for socioeconomic effects) forcings from 2015 to 2100[49]. Considerable consistency was observed between SSP5-8.5 and RCP85 scenarios[50], enhancing confidence in comparing the results from CESM-H and CMIP6. In addition, 14 CMIP6 simulations with a biogeochemistry component were utilized in the analysis of primary productivity in the EBUSs (Supplementary Table 1).

### Validation datasets
The Estimating the Circulation and Climate of the Ocean, Phase II (ECCO2) reanalysis product[51] was utilized for the verification of the CESM-H. ECCO2 provides results stored as a 3-day mean field on a 0.25° horizontal grid with 50 vertical levels, covering the period from 1992 to the present. To facilitate comparison with CESM-H, the 3-day mean vertical velocity at 50 meters spanning from 1992 to 2022 was linearly interpolated to a daily resolution.

The observed vertically integrated primary productivity dataset is an experimental dataset harvested from the Pacific Fisheries Environmental Laboratory. The daily file is derived from satellite-derived measurements of chlorophyll-a, incident visible surface irradiance, and sea surface temperature[52]. The monthly outputs, comprising a composite of the one-day files, have been available from January 2003 to the present, with a spatial resolution of approximately 0.05°.

## Region selection

In terms of seasonality, each EBUS is divided into three regions: the poleward region (-P), the central region (-C), and the equatorward region (-E). The poleward regions, located at high latitudes in each EBUS, experience a pronounced seasonal upwelling, with stronger upwelling predominantly occurring during summer and weaker upwelling or downwelling observed in winter. The equatorward regions, situated at low latitudes in each EBUS, also exhibit a significant seasonal upwelling. However, the upwelling reaches its peak during summer and weakens or even shifts to downwelling during winter, in contrast to the poleward regions. The central regions, positioned between the poleward and equatorward regions, exhibit a strong permanent upwelling but display limited seasonal variability across all EBUSs.

According to the aforementioned categorization principle, the CanCS is divided into three regions: the poleward region (CanCS-P), situated along the coast of the Iberian Peninsula (36°N–43°N), the central region (CanCS-C), encompassing latitudes ranging from 19°N-36°N, and the equatorward region (CanCS-E), traditionally referred to as the Mauritania-Senegalese upwelling region, spanning from 12°N to 19°N. Similarly, the HCS located in the South Pacific is also divided into three regions: HCS-E (5°S–15°S), HCS-C (15°S–36°S), and HCS-P (36°S–44°S). Likewise, the BCS situated in the South Atlantic is categorized as BCS-E (15°S-22°S), BCS-C (22°S-29°S), and BCS-P (29°S-33°S). The exception lies in the CalCS, where no winter-intensified seasonal upwelling was observed in its equatorward areas. Consequently, the CalCS region is divided solely into two regions: the central region (CalCS-C) spanning from 30°N to 40°N and the poleward region (CalCS-P) ranging from 40°N-48°N.

## Upwelling timing and intensity

The upwelling timing and intensity were defined based on the upwelling index (UI), computed as the volume of upwelled water at 50 m within ~200 km (20 model grids) from the coast[53].

$$\text{UI}(t) = \iint w(x, y, t) dx dy \qquad (1)$$

where $w$ is the vertical velocity. It should be noted that the regions situated ~200 km from the coast encompass the coastal upwelling and a substantial portion of the upwelling driven by offshore wind stress curl, both of which have implications for the EBUSs' ecosystem. It is worth noting that the mixed layer varies spatially and seasonally, and thus selecting a fixed depth as representative of the mixed layer may oversimplify its complexity. However, using a spatially varying depth poses challenges in closing the transport budget within the EBUSs[27], thereby complicating the analysis of the underlying dynamics governing upwelling changes. As a compromise, we opt for a fixed depth to represent the mixed layer in this study. Nevertheless, sensitivity tests show that minor depth changes do not have a substantial impact on the results in this study, leading support to the validity of the major conclusions.

The onset of the upwelling season is the date on which the UI turns negative to positive; similarly, the termination of the upwelling season is when the UI turns positive to negative. The duration of the upwelling season can be calculated by determining the total number of days between the onset and termination dates. However, since the transition of UI could be influenced by high-frequency fluctuations[54], we have applied a 1-year 4$^{\text{th}}$-order Savitzky-Golay filter[55] to remove the interseason signals of UI before the detection of onset and termination dates. The upwelling intensity is defined as the total volume of upwelled water during the upwelling season:

$$\text{Upwelling intensity} = \int_{onset}^{termination} \text{UI}(t) dt \qquad (2)$$

Both alterations in the duration of upwelling season and vertical velocity at a depth of 50 m have the potential to modify the the upwelling intensity. To assess the impact of velocity changes on upwelling intensity changes, we recalculated the velocity-induced upwelling intensity by keeping the onset and termination dates fixed at their climatological values (1920-1949). Consequently, the duration-induced upwelling intensity is derived as the differences between the total and velocity-induced upwelling intensity.

## Shifting definition in monthly data

The upwelling timing derived from monthly outputs is similar to the daily CESM-H method, albeit with an additional condition. To quantify the changes in monthly models' outputs over time, an epoch difference was calculated from two 30-year periods, spanning 1920-1949 and 2071-2100. Changes exceeding a month would be reflected by the models directly. However, the coarse temporal resolution of models could induce a constant timing of upwelling under greenhouse warming, as it is unable to capture changes within a few days. Hence, following Brady et al.[11]., a positive linear trend of UI at the transition month is regarded as an advancement in the onset or delay in the termination. Similarly, it can be inferred that a negative trend of UI during the transition month reflects a later onset or an earlier termination. The changes in duration are determined by the linear trend of the sum of UI at the onset month and termination month, with a positive depicting an extending duration.

## Trend analysis

The linear trends and their standard errors were obtained by regressing the time series of each upwelling metric against time using the Cochrane-Orcutt (C-O) method[56]. This method is employed to address the autocorrelation within the time series. The model for time series of a given quantity $\theta(t)$ is expressed as $\theta(t) = \beta_0 + \beta_1 t + \epsilon(t)$, where the intercept ($\beta_0$) and linear trend ($\beta_1$) can be easily computed based on the ordinary least-squares method. Here, $\epsilon(t)$ represents a stationary stochastic process with a zero mean. The standard error of the linear trend was computed using the C-O method, assuming that the autocorrelation in $\epsilon(t)$ can be approximated as a first-order autoregressive process.

## Decomposition of upwelling

The UI contributed by the wind-induced Ekman transport (UI$_e$) can be derived based on horizontal mass-flux divergence induced by Ekman transport[27]:

$$\text{UI}_e = \frac{1}{\rho_0 f} \oint \boldsymbol{\tau} \times \boldsymbol{k} \qquad (3)$$

where $\rho_0$ is the reference seawater density, $f$ is the Coriolis parameter varying with latitude, $\boldsymbol{\tau} = (\tau_x, \tau_y)$ is the surface wind stress, and $\boldsymbol{k}$ is the unit vector in vertical direction.

Similarly, the upwelling intensity contributed by geostrophic transport (UI$_g$) can be computed as the horizontal mass-flux divergence caused by geostrophic transport[16]:

$$\text{UI}_g = \int_{50\,m}^0 \oint \nabla \cdot \boldsymbol{u_g} dz$$
$$\boldsymbol{u_g} = -\frac{1}{\rho_0 f} \nabla p \times \boldsymbol{k} \qquad (4)$$

where $\boldsymbol{u_g}$ is the geostrophic velocity, $p$ is the seawater pressure computed according to the hydrostatic approximation $p = \rho_0 g \eta + \int_D^0 \rho g dz$, with $g$ the gravity acceleration, $\eta$ the sea surface height, and $\rho$ the seawater density.

## Computation of VNT

Nitrate input is an important component in nutrient supply as most of the phytoplankton growth was limited by the availability of inorganic nitrogen[57]. Thus, the vertical nutrient flux at 50 meters was calculated by multiplying the $w$ by the mole concentration of nitrate ($N$) at the same depth[58]. Afterward, the VNT, similar to the upwelling intensity definition, is defined as the integral vertical nutrient transport from the onset to the termination of the upwelling season.

$$\text{VNT} = \int_{onset}^{termination} w(x,y,t) \cdot N(x,y,t) dxdydt \tag{5}$$

To identify the effect of changes in upwelling and nutrients on VNT, we further decompose the VNT into components associated with the effects of upwelling ($\Delta w$), effects of nutrient ($\Delta N$), and their interaction effects:

$$\Delta w = \int_{onset}^{termination} w' \cdot \bar{N} \, dxdydt$$
$$\Delta N = \int_{onset}^{termination} \bar{w} \cdot N' \, dxdydt \tag{6}$$
$$\text{Interaction} = \int_{onset}^{termination} w' \cdot N' \, dxdydt$$

with the overbar denoting the climatological mean value and the prime is defined as perturbations from the climatological mean value.

## NPP during the upwelling season

The depth-integral primary organic carbon production over the full water column was regarded as the NPP in the ocean. Given that the euphotic layer depth is primarily shallower than 50 meters in the EBUSs[59], the full-depth-integral NPP could serve as a representative indicator for the upper-ocean NPP. Subsequently, the NPP during the upwelling season is calculated as the summation of NPP starting from the onset to the termination.

## Data availability

All data needed to evaluate the conclusions in the paper can be downloaded from the following links: CESM-H: https://ihesp.github.io/archive/products/ihesp-products/data-release/HIST_TNST/ocn/index.html and https://ihesp.github.io/archive/products/ihesp-products/data-release/RCP85_TNST/ocn/index.html; CMIP6 models: https://esgf-node.llnl.gov/search/cmip6/; ECCO2: https://ecco.jpl.nasa.gov/drive/files/ECCO2/cube92_latlon_quart_90S90N/; NPP: https://coastwatch.pfeg.noaa.gov/erddap/griddap/erdMH1ppmday.html. Source data of the main figures is provided with this paper. Source data are provided with this paper.

## Code availability

The DOI of the iHESP version of the CESM-H code is: https://doi.org/10.5281/zenodo.3637771.

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

## Acknowledgements

This work was supported by the National Natural Science Foundation of China (42325601), the Science and Technology Innovation Foundation of Laoshan Laboratory (Nos. LSKJ202202503), and the Taishan Scholar Funds (tsqn202306300). Computational resources were provided by the Sunway TaihuLight High-Performance Computer (Wuxi) and Laoshan Laboratory.

## Author contributions

T.D. conducted the analysis under S.W.'s instruction. S.W. proposed the central idea. Z.J. and L.W. led the research and organized the writing of the manuscript. T.D. and S.W. wrote the manuscript. C.Z. and B.Z. were involved in improving the manuscript.

## Competing interests

The authors declare no competing interests.
