## [Peer Review File · Nature Communications]

Future changes in coastal upwelling and biological production in eastern boundary upwelling systemsREVIEWER COMMENTS

Reviewer #1 (Remarks to the Author):

This is an interesting article related to climate change and Eastern Boundary Upwelling Systems (EBUSs). Throughout the study, the authors engage with hypotheses that are currently central to the EBUS community's research. One of the major challenges is linked to the coarse resolution of global climate models (both CMIP5 and CMIP6). In this context, the authors present a high-resolution global model with notable capabilities for evaluating upwellings. Additionally, they introduce an innovative methodology for accurately representing upwelling seasons. Finally, a key uncertainty regarding climate change in EBUSs is the fate of nutrients under climate change conditions. In this section, the authors address this uncertainty. The article is well-written and well-structured.

However, in my opinion, the results obtained in this study should undergo a thorough review before being published. My perspective is primarily supported by the high resolution demonstrated by the global model proposed by the authors. While one of the main uncertainties of EBUSs could be addressed with a high-resolution model, the authors negate this advantage by using averages at 200 km offshore, failing to adequately reflect the importance of regions on the shelf in EBUSs. Additionally, the authors do not accurately represent vertical transport in upwelling, as they use arbitrary depths for the four EBUSs with their respective latitudes and seasons. Alongside this, the authors present initial hypotheses about climate change and EBUSs, display relevant results (which they do not discuss) regarding the increase or decrease in upwelling but do not discuss or investigate which of the initial hypotheses align with their findings. Moreover, they introduce in the discussion the relationship of nutrients with stratification, when no results of oceanic stratification have been presented. Finally, the nutrient results are solely studied based on the CMIP6 global models, diminishing the significant advantage offered by the high-resolution model.

Therefore, I suggest that the authors undertake a comprehensive analysis of the comments I have provided below. Thus, the article should be resubmitted after a major revision.

Detailed comments:

L 31. Add that the transport is vertical towards the surface.

L 36 In Fig. 1 and subsequently in the rest of the sample analysis, the four EBUSs are shown. However, in the C-ICS, the authors present a domain cut off at approximately 26°N, where upwelling is most intense throughout the year (21°N -26°N; Vázquez et al. 2022), and the southernmost region (Mauritania-Senegalese) does not appear, which is one of the most intense in the C-ICS (Sylla et al. 2019). Additionally, the P-CCS region presents annual permanent upwelling from 5°S to 40°S, where it becomes seasonal as shown by the authors. In this case, the authors decide not to conduct the analysis from 17°S to 40°S (Kämpf and Chapman, 2016). Therefore, I

propose that the authors adjust the domains well to have a more exhaustive examination of their results and subsequent conclusions.

Vázquez, R., Parras-Berrocal, I., Cabos, W., Sein, DV., Mañanes, R. and Izquierdo, A. (2022). Assessment of the canary current upwelling system in a regionally coupled climate model. *Clim. Dyn.* 58, 69–85. <https://doi.org/10.1007/s00382-021-05890-x>.

Sylla, A., Mignot, J., Capet, X., Gaye, A.T., 2019. Weakening of the Senegalo–Mauritanian upwelling system under climate change. *Clim. Dyn.* 53 (7–8), 4447–4473. <https://doi.org/10.1007/s00382-01904797-y>.

Kämpf, J., Chapman, P. (2016). The Peruvian-Chilean Coastal Upwelling System. In: *Upwelling Systems of the World*. Springer, Cham. https://doi.org/10.1007/978-3-319-42524-5_5

L41-42 The reference to the lack of observational data is from 1973. Observational systems have advanced since then. Please, look for a more recent reference.

L43 Since the authors mention Bakun's hypothesis, please mention to the second hypothesis proposed in EBUSs's environment by Rykaczewki et al. (2015).

Rykaczewski, R.R., Dunne, J.P., Sydeman, W.J., Garcia-Reyes, M., Black, B.A., Bograd, S. J., 2015. Poleward displacement of coastal upwelling-favorable winds in the ocean's eastern boundary currents through the 21st century. *Geophys. Res. Lett.* 42, 6424–6431. <https://doi.org/10.1002/2015GL064694>.

L48-50 Neither of the two provided references is related to the author's proposed statement. I suggest to the authors to search for information and modify the citations. Here's a suggestion in this regard:

Wang, D.W., Gouhier, T.C., Menge, B.A., Ganguly, A.R., 2015. Intensification and spatial homogenization of coastal upwelling under climate change. *Nature* 518. <https://doi.org/10.1038/nature14235>

L68-69 What is the vertical resolution of the model? Please, also specify if this resolution is constant in the vertical and what is the thickness of each layer (especially in the first 200 m of the water column).

L79-84 It would help the understanding of the manuscript if you change austral and boreal summer-winter to the respective months they refer to (e.g., JJA – June-July-August).

L79 Please check the units for Figure 1c-f, as if the UI defined in the methodology is used, the units should be $m^3 s^{-1}$.

L89 I have a question about the comparison period between the model and the reanalysis. If the

model is driven under the RCP8.5 scenario, this simulation begins in 2005. Therefore, using the period 1992-2022 averages both historical model data and RCP8.5 scenario data. Wouldn't it be a better analysis to compare the period, for example, from 1992 to 2005?

L89 Regarding the reanalysis, ECCO2 has a horizontal resolution of $0.25^{\circ} \times 0.25^{\circ}$, which is coarse for comparison with an ocean model of such high resolution as CESM-H. I propose that you perform a comparison with the GLORYS model from Copernicus, which has a resolution of $0.083^{\circ} \times 0.083^{\circ}$ and may better align with the model.

Drévillon M, Regnier C, Lellouche JM, Garric G, Bricaud C, Hernandez O (2018) CMEMS-GLO-QUID-001-030, 1.2 edn. E.U.Copernicus Marine Service Information [Online]. <https://resources.marine.copernicus.eu/documents/QUID/CMEMS-GLOQUID-001-030.pdf>

L87-91 If I am not mistaken, the model overestimates the UI compared to the reanalysis data (supplementary Fig. 1 and Fig. 1). Please add a brief sentence discussing this fact. Although the differences are within an acceptable range, it is important to note in the manuscript to encourage the modelling community to strive for improvements in the future.

L94 I find the methodology proposed by the authors very interesting. However, I have a question about how you calculate the onset and termination of upwelling. Typically, the change from positive to negative in upwelling sign can be influenced by local wind patterns or even mesoscale processes from one day to the next. Have the authors applied any filters or analyses to calculate the average sign change over several days? Or have you simply calculated the sign change from one day to the next, and this does not vary from onset to termination?

L95 I have a question about the choice of poleward regions. In this analysis, the Benguela upwelling does not appear as the authors have not subdivided this system. Why? The BCS has the same latitudinal extent as the California upwelling, and the seasonal latitude-dependent difference is distinguishable in Fig. 1f. For example, it could be divided at 27°S . In this regard, please also include the BCS in the rest of the analyses, such as in Fig. 2.

L95. Add information to the graph to clarify that the y-axis of Fig. 2a-c represents dates.

L117. The study is conducted with the CESM-H model forced by the RCP8.5 scenario from CMIP5, but the comparison with global models is from CMIP6 (it is not mentioned in the text whether it is under the SSP5-8.5 scenario). Are there differences between the different scenarios? Add a sentence to the text or methodology discussing this.

L133 Change Pacific to Atlantic

L140-146 Please add the Benguela subdivision to compare whether it also increases, as in the Chile region.

L169-173 I do not believe this study provides results to claim that it can assess local nutrient

effects. By averaging over hundreds of kilometers both latitudinally and longitudinally, the ability to evaluate local effects is compromised (see Lovecchio et al. 2017, 2018; Santana-Falcón et al. 2020). Please avoid mentioning that local changes in nutrients are being assessed.

Lovecchio E, Gruber N, Münnich M, Lachkar Z (2017) On the longrange offshore transport of organic carbon from the Canary Upwelling System to the open North Atlantic. *Biogeosciences* 14(13):3337–3369. <https://doi.org/10.3929/ethz-b-000190480>

Lovecchio E, Gruber N, Münnich M (2018) Mesoscale contribution to the long-range offshore transport of organic carbon from the Canary Upwelling System to the open North Atlantic. *Biogeosciences* 15:5061–5091. <https://doi.org/10.5194/bg-15-5061-2018>

Santana-Falcón Y, Mason E, Arístegui J (2020) Offshore transport of organic carbon by upwelling filaments in the Canary Current System. *Prog Oceanogr* 184(April):102322. <https://doi.org/10.1016/j.pocean.2020.102322>

L173-174 I understand that the CESM-H model lacks biogeochemistry, so the authors use the global CMIP6 models to test the hypothesis. Although the results obtained previously by CMIP6 models could be compared with those of CESM-H, no evaluation of the models has been provided to see if they resolve latitudinal upwelling as in the case of CESM-H. In fact, there are different studies (Varela et al. 2022; Sylla et al. 2022) showing how CMIP6 models differ from observations. Therefore, I believe this section provides useful information that complements previous results and helps take a first step regarding the biogeochemical aspect of upwelling. However, a paragraph should be added discussing the deficiencies of CMIP6 models in reproducing EBUSs, primarily associated with the lack of resolution.

Varela, R.; DeCastro, M.; Rodriguez-Diaz, L.; Dias, J.M.; Gómez-Gesteira, M. Examining the Ability of CMIP6 Models to Reproduce the Upwelling SST Imprint in the Eastern Boundary Upwelling Systems. *J. Mar. Sci. Eng.* 2022, 10, 1970. <https://doi.org/10.3390/jmse10121970>

Sylla, A., Sanchez Gomez, E., Mignot, J., and López-Parages, J.: Impact of increased resolution on the representation of the Canary upwelling system in climate models, *Geosci. Model Dev.*, 15, 8245–8267, <https://doi.org/10.5194/gmd-15-8245-2022>, 2022.

L185-186 The mentioned section in the methods does not exist.

L216-219 Do you compare the Mixed Layer Depth (MLD) and nutrients based on averages from the areas shown in Fig 1? If so, I believe changes in MLD may be inadequately represented as it can be highly variable from one grid cell to another. Additionally, the authors do not assess stratification at any point. I suggest they calculate the Brunt-Väisälä frequency as an indicator of stratification changes (see Vázquez et al. 2023)

Vázquez, R., Parras-Berrocal, I. M., Koseki, S., Cabos, W., Sein, D. V., & Izquierdo, A. (2023). Seasonality of coastal upwelling trends in the Mauritania-Senegalese region under RCP8.5 climate

change scenario. *Science of the Total Environment*, 898, 166391.

Major comments:

The first question I have regarding the results in this work is related to the use of model vertical velocity outputs, as well as the use of an arbitrary depth (50 m) as the most representative of upwelling. I would like the authors to justify why they chose to use 50 m. In studies on upwelling systems, it has been agreed upon that there is significant variability both latitudinally and seasonally in the upwelling source depth. Using 50 m as a criterion oversimplifies the complexity of EBUSs. In this regard, using the MLD as the depth at each latitude and for each season would be more appropriate. However, estimating the appropriate depth remains highly complex. In fact, Jacox et al. (2018) proposes another method to assess upwelling volume that is superior to model vertical velocity outputs (Marchesiello and Estrade, 2010; Rossi et al. 2013; Oerder et al. 2015). In this context, I include the paragraph that describes this issue:

“However, we do not use modeled vertical velocities to construct the index for several reasons: First, the calculation of upwelling using modeled vertical velocity is not straightforward. One must first define a representative depth at which vertical velocity is extracted (e.g., the MLD), which will be different for each grid cell, making it difficult to close the transport budget. For example, when the MLD differs between adjacent grid cells, water can enter/exit the surface mixed layer horizontally and that component of the transport will be missed from the upwelling/downwelling estimate.”

Therefore, I would like to suggest to the authors that they reconsider the use of the MLD as the depth criterion and discuss the use of vertical velocity versus the wind field.

Marchesiello, P., and P. Estrade (2010), Upwelling limitation by onshore geostrophic flow, *J. Mar. Res.*, 68, 37–62

Rossi, V., Feng, M., Pattiaratchi, C., Roughan, M., & Waite, A. M. (2013). On the factors influencing the development of sporadic upwelling in the Leeuwin Current system. *Journal of Geophysical Research: Oceans*, 118(7), 3608-3621.

Oerder, V., Colas, F., Echevin, V., Codron, F., Tam, J., Belmadani, A., 2015. Peru-Chile upwelling dynamics under climate change. *J. Geophys. Res. Oceans* 120 (2), 1152–1172.
<https://doi.org/10.1002/2014JC0102>, 99.

Jacox, M.G., Edwards, C.A., Hazen, E.L., Bograd, S.J., 2018. Coastal upwelling revisited: Ekman, Bakun, and improved upwelling indices for the U.S. West Coast. *J. Geophys. Res. Oceans* 123, 7332–7350. <https://doi.org/10.1029/2018JC014187>.

The second concern is related to the domain chosen for the zonal average. In this regard, the authors average over 200 km offshore and mention that this is most appropriate for capturing both Ekman transport-associated upwelling and upwelling associated with Ekman transport divergence.

Averaging over such a wide portion of the coast may obscure the upwelling transport. Perhaps with low-resolution global models, obtaining a 200 km average is necessary. However, with a model of such high resolution as proposed in this study, the authors should consider narrowing this region to a maximum of 100 km. This would allow for a more appropriate assessment of upwelling, where mesoscale processes play a significant role. Therefore, I propose that the authors conduct the analysis by limiting the average to the first 100 km of the coast. In fact, in the methodology, the authors mention the work of Du et al. (2023), who also conducted a study with an average over the first 50 km offshore. Moreover, in Fig. 1a of the manuscript, the vertical velocity associated with upwelling is closely confined to the coast.

Reviewer #2 (Remarks to the Author):

This paper is a model-based study (CESM-H, CMIP6, ECCO2) aimed at understanding coastal upwelling variability in the context of the climate change. While the authors have undertaken significant efforts to comprehend the seasonal progression and upwelling intensity in the EBUSs, there are concerns regarding certain processes and model biases that prevent me from approving its publication.

Major concerns:

- 1) Recently, it has been pointed out that the geostrophic flow may play a significant role in controlling the upwelling trends under the climate change (Jing et al., 2023). The authors should include an explanation regarding whether and why it is not being considered in this study. Could the results be the same if the geostrophic flow is considered?
- 2) For the computation of the upwelling timing and intensity, the vertical velocity at 50 m depth has been considered. Could the results be different using another depth? (Note that Xiu et al. (2018) shows that, at least for CCS, there are three major drivers of the vertical velocity changes which are depth-dependent: alongshore winds, wind-curl and mesoscale activity)
- 3) How do the VNT and NPP appear under historical period compared to observational data (e.g., 2003-2022)? Most of the models exhibit a significant bias in the EBUSs, particularly with the biogeochemical variables. Can we have confidence in the results considering these biases?

Minor points:

- Line 111-114: Is this valid for Chilean and Peruvian region? (as far as it is known, these two regions have different upwelling regimes, which one being permanent and the other seasonal)
- Line 235 Fig.1: why are figures 1c and 1d composed of two subfigures? (Should they be like 1f?)
- Line 515 Supp. Fig.2b and 2c: the curves in these figures represent the UI only for one of the initially selected subdivisions for each upwelling system (as shown in figure 1)?
- Line 547 Supp. Table 1: Include units on the table

References:

Jing, Z., Wang, S., Wu, L. et al. Geostrophic flows control future changes of oceanic eastern boundary upwelling. *Nat. Clim. Chang.* 13, 148–154 (2023). <https://doi.org/10.1038/s41558-022-01588-y>

Xiu, P., Chai, F., Curchitser, E.N. et al. Future changes in coastal upwelling ecosystems with global warming: The case of the California Current System. *Sci Rep* 8, 2866 (2018). <https://doi.org/10.1038/s41598-018-21247-7>

Reviewer #3 (Remarks to the Author):

Title: Review of ‘Future changes in coastal upwelling and biological production in eastern boundary upwelling systems’

The paper investigates future changes in the duration and intensity of upwelling in the eastern boundary regions and its potential consequences on primary production. Utilizing simulations from CMIP6 models and high-resolution CESM, the authors showed that the duration of upwelling is projected to be longer under high-emission scenario. But changes in the upwelling intensity show region-specific features. The authors further explore the implications of these intensity changes on net primary production. While the results are interesting and seem to be scientifically sound, this paper still needs substantial work before it can be published. I summarize my concerns and comments below.

Major concerns:

1) Refinement of introduction. This paper introduces a novel investigation into changes in upwelling duration, a distinctive aspect that warrants attention. However, it is crucial to incorporate relevant findings from existing literature to enrich the contextual background. For instance, Wang et al. (2015), utilizing CMIP5 models, observed an increased duration of upwelling in the high latitudes of major EBUSs, with differing trends in low latitudes. Summarizing these findings explicitly in the introduction is essential to provide readers with a comprehensive understanding of the advancements in this field. In addition, the introduction of future changes in net primary production only discussed ocean stratification and upwelling intensity. However, nutrients in the source water can also impact future changes in net primary production (Rykaczewski and Dunne, 2010). Please make the introduction more comprehensive.

2) Refinement of Chile and Canary-Iberian upwelling systems definition. The definition of the Chile and Canary-Iberian upwelling systems requires careful consideration based on the results presented in Figure 1a. The depicted large upward vertical velocity extends beyond the designated boxes in the Chile and Canary regions, reaching substantially equatorward. Notably, the surface wind in the Chile box is perpendicular to the coast, which may not accurately represent the typical Chilean upwelling regions. Additionally, it is observed that stronger coast-parallel winds (as shown in Figure 1a) and higher net primary production (as indicated in Figure 1b) occur south of the defined Canary box.

In addition, if authors define upwelling regions based on vertical velocity, Figure 1a demonstrates

that positive vertical velocity has very narrow zonal extents. This suggests that selecting 200 km away from the coast will include regions with negative vertical velocity. Given that upwelling timing and intensity calculations rely on box averages, the selection of these boxes has the potential to influence the results of future changes in upwelling intensity and duration. To ensure the accuracy of the study's findings, a reconsideration of the chosen boxes for defining the Chile and Canary-Iberian upwelling systems is recommended. This adjustment should aim to better capture the representative characteristics of these upwelling regions and minimize potential biases in the assessment of future changes in upwelling dynamics.

3) Lack of mechanisms in upwelling changes. One notable limitation in the current study lies in the absence of a detailed exploration of the underlying mechanisms driving changes in the duration and intensity of upwelling. This omission weakens the novelty of the study, especially when compared to the work of Wang et al. (2015), who utilized CMIP5 models and already delved into discussions on these changes, demonstrating general consistency with the results presented in this paper. To strengthen the scientific contribution and novelty of this study, it is imperative to incorporate a thorough examination of the mechanisms influencing the observed changes in upwelling duration and intensity.

4) Refinement of methodology concerning shifting definition in monthly data. A critical aspect requiring attention in the methodology is the method of shifting definition employed in analyzing upwelling duration based on monthly data, utilizing the epoch difference technique (Brady et al., 2017). While this method proves effective for studying the emergence of anthropogenic impacts on upwelling intensity, its applicability to upwelling duration analysis is questionable. As shown in Figure 2a-c, the onset and termination time suffer strong interannual/decadal variabilities. From 1920 to 2010, onset time varies from 01-15 to 03-01 in northern CalCS and Iberian system, and from 07-15 to 08-15 in Chilean system. Such variability suggests that changes in onset time using a 30-year epoch difference can be influenced by internal variability rather than anthropogenic forcing, especially when dealing with monthly data. The same problem arises from the changes in termination time. This implies that the comparison between CMIP6 and CESM-H, based on this methodology, may not be a fair one. To address this issue and ensure a more robust comparison, it is advisable to focus on confirming the role of surface wind in influencing changes in upwelling duration. Subsequently, employing daily wind data to estimate duration in CMIP6 models, as demonstrated by Wang et al. (2015), can provide a more accurate representation of the temporal dynamics of upwelling.

Minor concerns:

1) Descriptions of upwelling regions are incomplete. Firstly, the authors did not explicitly define Northern CalCS, Iberian, and Chilean regions, potentially confusing readers. Secondly, the authors should describe methods to calculate the trend errors of onset and termination time that are shown in Figure 2a-c. Thirdly, the term "equatorward regions" is used without a precise definition.

2) What are the x-axis and numbers -3, -2, ..., 2, 3 in Figure 2d-f? Authors should describe it in the caption.

3) The definition of upwelling intensity in the study is based on the time and spatial integral of vertical velocity at 50 m, with onset and termination times determined using monthly data. However, an important aspect requiring clarification is the attribution of changes in upwelling intensity. The study does not explicitly specify whether alterations in upwelling intensity are primarily driven by variations in upwelling duration, vertical velocity at 50 m, or a combination of

both factors.

4) The authors consistently express the agreement percentages between CMIP6 simulations and CESM-H, but it is crucial to note that not all CMIP6 simulations were utilized in the paper. To enhance precision and clarity in reporting, it is recommended to present the results in a format that specifies the number of simulations in agreement, such as "xx out of xx simulations." This format provides a more accurate representation of the agreement statistics, ensuring that readers have a clear understanding of the scope and basis for the reported percentages.

5) CESM-H models used in this study do not include biogeochemistry components, which means that nitrate in CESM-H is a passive tracer. Given this, it is not surprising that changes in vertical velocity dominate changes in vertical nitrate fluxes, as no additional sources and sinks of nitrate are considered in the future. To deepen the understanding of the biogeochemical dynamics, it is recommended to explore whether, in CMIP6 models with biogeochemistry components, changes in vertical nitrate flux are similarly dominated by changes in vertical velocity. This comparison will shed light on the role of biogeochemistry in influencing vertical nitrate flux patterns. In addition, total vertical nitrate flux calculated using monthly data does not include the component induced by submonthly variabilities (equivalent to eddy component). The eddy flux is resolved by CESM-H but parameterized in 1° or coarser CMIP6 models. Therefore, it is advisable to quantify the eddy fluxes, leveraging the advantages of CESM-H, to provide a more comprehensive understanding of the nuances in vertical nitrate fluxes and their implications.

Specific comments:

L34: EBUSs generally stands for eastern boundary upwelling systems, not coastal upwelling systems.

L41-42: This statement is not accurate. The reason for using wind to estimate upwelling is because upwelling is mainly wind-driven, not because sparsity of oceanic observations.

L47-48: Previous studies showed that projected changes in wind and upwelling both show inconsistency across different models. But in one certain model, if it is not wind determining future changes in upwelling intensity, what are the other factors? Please clarify it here.

Figure 1a: vertical velocity at which depth?

L76: 'Consistent with the existing theoretical arguments and simulations', please add references here.

L83: The maximum between 14°S-17°S in P-CCS and 27°S-28°S in BCS occurs in the early austral spring. Please precisely define poleward and equatorward regions for P-CCS and BCS.

L86-87: Because duration of upwelling is the major focus of this paper, it will be more comprehensive to provide a figure including duration as a function of latitude in each upwelling system. It is difficult to follow the authors' argument without any figures like that.

L104: 'all EBUSs' means Northern CalCS, Iberian, and Chilean regions or regions defined by boxes in Figure 1a?

L129: Authors divided EBUSs into poleward and equatorward regions, which show different changes in duration (Figure 2 and Supplementary Figure 2). 'The consistent changes across all EBUSs ...' is confusing.

L123-124: 75% and 73% are equivalent to how much out of how much?

L133: 'North Atlantic'?

L134-135: Canary and Iberian regions are not defined.

L138: north and south portions of CalCS are not defined.

L140-141: As shown in Figure 1, the Benguela system extends from 33°S to 15°S. Authors cannot simply group it into equatorward systems.

L157, 159, 161: Please add actual numbers (xx out of xx) after the percentage because you are not using all models that participated in CMIP6.

L336: Should be 'Brady et al.'

References:

Wang, D., Gouhier, T. C., Menge, B. A., & Ganguly, A. R. (2015). Intensification and spatial homogenization of coastal upwelling under climate change. *Nature*, 518(7539), 390-394.

Brady, R. X., Alexander, M. A., Lovenduski, N. S., & Rykaczewski, R. R. (2017). Emergent anthropogenic trends in California Current upwelling. *Geophysical Research Letters*, 44(10), 5044-5052.

Rykaczewski, R. R., & Dunne, J. P. (2010). Enhanced nutrient supply to the California Current Ecosystem with global warming and increased stratification in an earth system model. *Geophysical Research Letters*, 37(21).

Reply to the first reviewer

We are very grateful to you for your time in carefully reading our manuscript and providing helpful comments that make our manuscript better. We have carefully considered each of your comments (in blue) and revised the manuscript accordingly. Please find our response (in black) to your comments below.

Reviewer #1 (Remarks to the Author):

This is an interesting article related to climate change and Eastern Boundary Upwelling Systems (EBUSs). Throughout the study, the authors engage with hypotheses that are currently central to the EBUS community's research. One of the major challenges is linked to the coarse resolution of global climate models (both CMIP5 and CMIP6). In this context, the authors present a high-resolution global model with notable capabilities for evaluating upwellings. Additionally, they introduce an innovative methodology for accurately representing upwelling seasons. Finally, a key uncertainty regarding climate change in EBUSs is the fate of nutrients under climate change conditions. In this section, the authors address this uncertainty. The article is well-written and well-structured.

However, in my opinion, the results obtained in this study should undergo a thorough review before being published. My perspective is primarily supported by the high resolution demonstrated by the global model proposed by the authors. While one of the main uncertainties of EBUSs could be addressed with a high-resolution model, the authors negate this advantage by using averages at 200 km offshore, failing to adequately reflect the importance of regions on the shelf in EBUSs. Additionally, the authors do not accurately represent vertical transport in upwelling, as they use arbitrary depths for the four EBUSs with their respective latitudes and seasons. Alongside this, the authors present initial hypotheses about climate change and EBUSs, display relevant results (which they do not discuss) regarding the increase or decrease in upwelling but do not discuss or investigate which of the initial hypotheses align with

their findings. Moreover, they introduce in the discussion the relationship of nutrients with stratification, when no results of oceanic stratification have been presented. Finally, the nutrient results are solely studied based on the CMIP6 global models, diminishing the significant advantage offered by the high-resolution model.

Therefore, I suggest that the authors undertake a comprehensive analysis of the comments I have provided below. Thus, the article should be resubmitted after a major revision.

Detailed comments:

L 31. Add that the transport is vertical towards the surface.

We have revised it as “... facilitates the vertical transport of nutrient-rich water to the surface.”

L 36 In Fig. 1 and subsequently in the rest of the sample analysis, the four EBUSs are shown. However, in the C-ICS, the authors present a domain cut off at approximately 26°N, where upwelling is most intense throughout the year (21°N -26°N; Vázquez et al. 2022), and the southernmost region (Mauritania-Senegalese) does not appear, which is one of the most intense in the C-ICS (Sylla et al. 2019). Additionally, the HCS region presents annual permanent upwelling from 5°S to 40°S, where it becomes seasonal as shown by the authors. In this case, the authors decide not to conduct the analysis from 17°S to 40°S (Kämpf and Chapman, 2016). Therefore, I propose that the authors adjust the domains well to have a more exhaustive examination of their results and subsequent conclusions.

Thanks for your comment. In the revised manuscript, we have separately subdivided the Canary Current, Humboldt Current and Benguela Current systems into three regions (equatorward region, central region and poleward region) based on differences in upwelling seasonality. In this case, the Mauritania-Senegalese is encompassed within the equatorward region of the Canary Current system and is analyzed separately. Similarly, the permanent upwelling region from 15°S to 40°S is encompassed within the central Humboldt Current system and also analyzed in the revised manuscript. The

subsequent results and conclusions were revised accordingly, with the elaborate definition of the regions added in line 422-443.

L41-42 The reference to the lack of observational data is from 1973. Observational systems have advanced since then. Please, look for a more recent reference.

Thanks for your comments. Following other reviewers' comment, this sentence has been removed.

L43 Since the authors mention Bakun's hypothesis, please mention to the second hypothesis proposed in EBUSs's environment by Rykaczewski et al. (2015).

Thanks for your comments, we have added the hypothesis by Rykaczewski et al. (2015) to the introduction, see in line 45-48

“However, recent studies have underscored the impact of poleward migration of the high-pressure systems on future changes in coastal upwelling, projecting an intensified upwelling in higher latitudes and weakened upwelling in lower latitudes (Rykaczewski et al., 2015).”

L48-50 Neither of the two provided references is related to the author's proposed statement. I suggest to the authors to search for information and modify the citations. Here's a suggestion in this regard:

Wang, D.W., Gouhier, T.C., Menge, B.A., Ganguly, A.R., 2015. Intensification and spatial homogenization of coastal upwelling under climate change. *Nature* 518. <https://doi.org/10.1038/nature14235>

Thanks for your suggestion, the reference has been changed to:

García-Reyes, M., Sydeman, W. J., Schoeman, D. S., Rykaczewski, R. R., Black, B. A., Smit, A. J., & Bograd, S. J. (2015). Under pressure: climate change, upwelling, and eastern boundary upwelling ecosystems. *Frontiers in Marine Science*, 2. Retrieved from <https://www.frontiersin.org/articles/10.3389/fmars.2015.00109>

Wang, D., Gouhier, T. C., Menge, B. A., & Ganguly, A. R. (2015). Intensification and spatial homogenization of coastal upwelling under climate change. *Nature*, 518(7539), 390–394. <https://doi.org/10.1038/nature14235>

L68-69 What is the vertical resolution of the model? Please, also specify if this resolution is constant in the vertical and what is the thickness of each layer (especially in the first 200 m of the water column).

Thanks for your comment. For the oceanic component of CESM-H, there are 62 levels in the vertical with increasing grid space from 10 m near the sea surface to 250 m near the bottom. In special, the thickness of each layer is identically set as 10 m in the upper 150 m, below which the thickness is gradually increase with depth. We have added the detailed information on the model in the revised manuscript, see lines 391-394.

L79-84 It would help the understanding of the manuscript if you change austral and boreal summer-winter to the respective months they refer to (e.g., JJA – June-July-August).

Thanks for your suggestion. Since we have changed the expressions of the upwelling seasonality in the revised manuscript, thus, the usage of austral/boreal summer/winter has been replaced to summer/winter. However, we have still incorporated the corresponding months for the summer and winter in the Northern and Southern Hemispheres into the revised manuscript, please see in line 91-92

L79 Please check the units for Figure 1c-f, as if the UI defined in the methodology is used, the units should be $m^3 s^{-1}$.

Sorry for our mistake, this has been revised. See Figure 1.

L89 I have a question about the comparison period between the model and the reanalysis. If the model is driven under the RCP8.5 scenario, this simulation begins in 2005. Therefore, using the period 1992-2022 averages both historical model data and RCP8.5 scenario data. Wouldn't it be a better analysis to compare the period, for example, from 1992 to 2005?

As reported in previous studies, the upwelling in the Eastern Boundary Upwelling Systems is influenced by natural variability and exhibits inherent multi-scale temporal variability. Therefore, selecting a longer period for comparing the simulated vertical velocity with observations would enable the removal of natural variability-induced influences. Traditionally, employing a minimum 30-year average is considered effective in mitigating the natural variability and widely used in climate studies (Ding et al., 2021; Rykaczewski et al., 2015). We pretty agree with you that employing the RCP simulation may introduce uncertainties when compared to the observations. However, considering that current CO₂ emissions estimates align closely with the RCP8.5 projection (Figure 1 in Peters et al., 2013), using the period 1992-2022 for model-reanalysis comparison is considered reasonable.

We have also performed a sensitivity analysis to evaluate the influence of selecting different time period for comparing the simulated vertical velocity with the observations. As shown in Fig. R1 and R2, the CESM-H also exhibit a high level of consistency with ECCO2 during 1992-2005, albeit with some minor quantitative variations.

Figure R1. Comparison between CESM-H (a-d) and ECCO2 (e-h) during 1992 to 2005.

Figure R2. Same as Figure R1 but for 1992 to 2022.

Ding, H., Alexander, M. A., & Jacox, M. G. (2021). Role of Geostrophic Currents in Future Changes of Coastal Upwelling in the California Current System. *Geophysical Research Letters*, 48(3), e2020GL090768. <https://doi.org/10.1029/2020GL090768>

Peters, G. P., Andrew, R. M., Boden, T., Canadell, J. G., Ciais, P., Le Quéré, C., et al. (2013). The challenge to keep global warming below 2 °C. *Nature Climate Change*, 3(1), 4–6. <https://doi.org/10.1038/nclimate1783>

L89 Regarding the reanalysis, ECCO2 has a horizontal resolution of 0.25°x0.25°, which is coarse for comparison with an ocean model of such high resolution as CESM-H. I propose that you perform a comparison with the GLORYS model from Copernicus, which has a resolution of 0.083°x0.083° and may better align with the model.

Thanks for your comment. The primary reason for utilizing ECCO2 for comparison is because its direct output of the vertical velocity (w) variable. While GLORYS has a higher spatial resolution, it lacks the w variable output. Nevertheless, we calculated w using the horizontal velocity derived from GLORYS via the continuity equation $w(z = -h) = -\int_{-h}^0 \left(\frac{\partial u}{\partial x} + \frac{\partial v}{\partial y} \right) dz$ (Tarry et al., 2022). The results demonstrate agreement with CESM-H both in pattern and magnitude (Fig. R3). The computation of the

continuity equation, however, is sensitive to the adopted deviational method, which may introduce certain uncertainties. Therefore, despite the higher spatial resolution of GLORYS, we choose ECCO2 as the validation products due to its direct output of w .

Figure R3. Same as Fig. 1c-f but for results derived from GLORYS.

Tarry, D. R., Ruiz, S., Johnston, T. M. S., Poulain, P.-M., Özgökmen, T., Centurioni, L. R., et al. (2022). Drifter Observations Reveal Intense Vertical Velocity in a Surface Ocean Front. *Geophysical Research Letters*, 49(18), e2022GL098969. <https://doi.org/10.1029/2022GL098969>

L87-91 If I am not mistaken, the model overestimates the UI compared to the reanalysis data (supplementary Fig. 1 and Fig. 1). Please add a brief sentence discussing this fact. Although the differences are within an acceptable range, it is important to note in the manuscript to encourage the modelling community to strive for improvements in the future.

Thanks for your advices. We fully agree with you that the simulated UI is overestimated by the CESM-H compared with the ECCO2 reanalysis. As noted in previous studies, the coastal upwelling zone simulated in oceanic models may be set by the parameterized physical or numerical horizontal mixing via their effects on the horizontal Ekman layer width. Further increasing the oceanic resolution would probably make the coastal upwelling stronger and narrower (Chang et al., 2023). Thus, the upwelling intensity in ECCO2 may be underestimated due to the coarse resolution of its oceanic model. Indeed, in GLORYS, which has a higher resolution, the simulated upwelling intensity is more comparable to that in the CESM-H (Fig. R3), leading further supports to our

hypothesis. However, even CESM-H is not fine enough to well resolve the coastal upwelling. We have cautioned readers about this issue. Please see lines 104-106.

Chang, P., Xu, G., Kurian, J., Small, R. J., Danabasoglu, G., Yeager, S., et al. (2023). Uncertain future of sustainable fisheries environment in eastern boundary upwelling zones under climate change. *Communications Earth & Environment*, 4(1), 1–9. <https://doi.org/10.1038/s43247-023-00681-0>

L94 I find the methodology proposed by the authors very interesting. However, I have a question about how you calculate the onset and termination of upwelling. Typically, the change from positive to negative in upwelling sign can be influenced by local wind patterns or even mesoscale processes from one day to the next. Have the authors applied any filters or analyses to calculate the average sign change over several days? Or have you simply calculated the sign change from one day to the next, and this does not vary from onset to termination?

As you said, the upwelling index (UI) sign could be influenced by local wind patterns or even mesoscale processes from one day to the next. Thus, it is difficult to detect the precise transition day from the downwelling to the upwelling. Here, we have applied a 1-year 4th-order Savitzky-Golay filter (Savitzky & Golay, 1964) to remove the inter-season signals of UI before the detection of onset and termination dates. This would help us to smooth the seasonal cycle and avoid the high-frequency fluctuation influences. In the revised manuscript, we have added this information to make the methodology clearer. Please see Line 462-465.

Savitzky, Abraham., & Golay, M. J. E. (1964). Smoothing and Differentiation of Data by Simplified Least Squares Procedures. *Analytical Chemistry*, 36(8), 1627–1639. <https://doi.org/10.1021/ac60214a047>

L95 I have a question about the choice of poleward regions. In this analysis, the Benguela upwelling does not appear as the authors have not subdivided this system. Why? The BCS has the same latitudinal extent as the California upwelling, and the seasonal latitude-dependent difference is distinguishable in Fig. 1f. For example, it

could be divided at 27°S. In this regard, please also include the BCS in the rest of the analyses, such as in Fig. 2.

Thanks for your comment. In the revised manuscript, we have subdivided the Benguela system into three regions (poleward (-P), central (-C), and equatorward (-E)) based on differences in upwelling seasonality, intensity as well as its potential dynamics. In particular, the BCS-E (15°S-22°S) is located at the poleward part of Benguela system, exhibiting a significant seasonal variability, with a greater magnitude during the winter and a lesser magnitude during the summer, respectively. The BCS-C region (22°S-29°S) exhibit a strong permanent upwelling but weak seasonal variability. As for the BCS-P region (29°S-34°S), situated at the southernmost tip of the African continent, also exhibits significant seasonal variability. However, the upwelling seasonal variability exhibits a greater magnitude during summer and a lesser magnitude during winter, which is in contrast to that in BCS-E.

Following the new subdivision method, we have included BCS in the rest of the analyses. However, different from other systems, the BCS region exhibits a permanent upwelling in every subdomain. Furthermore, the permanent upwelling in BCS is projected to continue through the 21st century, indicating non-anticipated changes in the timing and duration of the upwelling season. Therefore, we have subtracted the BCS region from the timing analysis as shown in Fig 2.

L95. Add information to the graph to clarify that the y-axis of Fig. 2a-c represents dates.

Thanks for your suggestion, we have added the information in the revised Fig. 2 and its caption.

L117. The study is conducted with the CESM-H model forced by the RCP8.5 scenario from CMIP5, but the comparison with global models is from CMIP6 (it is not mentioned in the text whether it is under the SSP5-8.5 scenario). Are there differences between the different scenarios? Add a sentence to the text or methodology discussing this.

Thanks for your comment. As you said, the CESM-H model is forced by the RCP8.5 scenario, while the CMIP6 is forced by the SSP5-8.5 scenario. Compared with the RCP8.5 scenario, the SSP5-8.5 scenario is also combined with the socioeconomic reasons. However, considerable consistency was observed between these two scenarios (Gulev et al., 2021), enhancing confidence in comparing the results from CESM-H with CMIP6. We have added this discussion in the revised manuscript, please see in line 402-407.

L133 Change Pacific to Atlantic

Revised.

L140-146 Please add the Benguela subdivision to compare whether it also increases, as in the Chile region.

Thanks for your comments. Following your advice, we have subdivided the Benguela system into three subdomains (See our reply to your comment L95). As shown in Fig. R4, all the subdomains in the Benguela are projected to experience an attenuation in upwelling intensity. However, the magnitude of the decreasing trends is highly dependent on latitudes. In particular, the equatorward region in BCS (BCS-E) is projected to experience a significant attenuation in upwelling intensity at a rate -12.4% per century. As for the central region in BCS (BCS-C), the upwelling intensity also exhibit a decreasing trend (-7.1% per century), but with a weaker magnitude than that in the BCS-E. In contrast, the BCS-P, located at the poleward portion of the South Atlantic, exhibits a very faint negative trend (-1.0% per century) in upwelling intensity. This decreasing trend with increasing latitudes is consisted with that in the poleward and central regions of HCS located at similar latitudes.

Different from other upwelling systems, the BCS region is located at a lower latitude due to the geographical limitation of African continent. Thus, the increasing intensity trend in the Chile region (renamed as HCS-P in the revised manuscript) could not be detected in the BCS region, due to its location in a higher latitude. However, based on the observed pattern of decreasing upwelling intensity in the BCS with latitudes, it can

be inferred that if the Africa continent extends to higher latitudes, there is a potential for a future increasing trend in upwelling intensity at higher latitudes.

This discussion has been added in the revised manuscript, please see in line 197-198.

Fig. R4 Projected changes in upwelling intensity in EBUSs.

L169-173 I do not believe this study provides results to claim that it can assess local nutrient effects. By averaging over hundreds of kilometers both latitudinally and longitudinally, the ability to evaluate local effects is compromised (see Lovecchio et al. 2017, 2018; Santana-Falc3n et al. 2020). Please avoid mentioning that local changes in nutrients are being assessed.

Lovecchio E, Gruber N, Munnich M, Lachkar Z (2017) On the longrange offshore transport of organic carbon from the Canary Upwelling System to the open North Atlantic. *Biogeosciences* 14(13):3337–3369. <https://doi.org/10.3929/ethz-b-000190480>

Lovecchio E, Gruber N, Munnich M (2018) Mesoscale contribution to the long-range offshore transport of organic carbon from the Canary Upwelling System to the open North Atlantic. *Biogeosciences* 15:5061–5091. <https://doi.org/10.5194/bg-15-5061-2018>

Santana-Falcón Y, Mason E, Arístegui J (2020) Offshore transport of organic carbon by upwelling filaments in the Canary Current System. *Prog Oceanogr* 184(April):102322. <https://doi.org/10.1016/j.pocean.2020.102322>

Sorry for our misleading statement. We have revised this statement in the revised manuscript to avoid misleading readers.

L173-174 I understand that the CESM-H model lacks biogeochemistry, so the authors use the global CMIP6 models to test the hypothesis. Although the results obtained previously by CMIP6 models could be compared with those of CESM-H, no evaluation of the models has been provided to see if they resolve latitudinal upwelling as in the case of CESM-H. In fact, there are different studies (Varela et al. 2022; Sylla et al. 2022) showing how CMIP6 models differ from observations. Therefore, I believe this section provides useful information that complements previous results and helps take a first step regarding the biogeochemical aspect of upwelling. However, a paragraph should be added discussing the deficiencies of CMIP6 models in reproducing EBUSs, primarily associated with the lack of resolution.

Thanks for your comments and useful references. As evidenced by previous studies, the accuracy of upwelling simulation is contingent upon on realistic representation of the cross-shore wind stress structure, seafloor topography, and oceanic mesoscale eddies, which all connect to the high-resolution in oceanic or atmospheric model configuration. Thus, using the low-resolution CMIP6 models would entail large uncertainties in projecting the future changes in upwelling intensity and associated biogeochemical changes. In the revised manuscript, we have cautioned readers about this issue. Please see Lines 303-306.

L185-186 The mentioned section in the methods does not exist.

Revised.

L216-219 Do you compare the Mixed Layer Depth (MLD) and nutrients based on averages from the areas shown in Fig 1? If so, I believe changes in MLD may be inadequately represented as it can be highly variable from one grid cell to another.

Additionally, the authors do not assess stratification at any point. I suggest they calculate the Brunt-Väisälä frequency as an indicator of stratification changes (see Vázquez et al. 2023)

Vázquez, R., Parras-Berrocal, I. M., Koseki, S., Cabos, W., Sein, D. V., & Izquierdo, A. (2023). Seasonality of coastal upwelling trends in the Mauritania-Senegalese region under RCP8.5 climate change scenario. *Science of the Total Environment*, 898, 166391.

Thanks for your comment and useful reference. Following the method in Vázquez et al. (2023), we have calculated the upper 150 m Brunt-Väisälä frequency as an indicator of stratification changes instead of the MLD in the revised manuscript. As shown in Fig R5, the stratification in major EBUSs exhibits pronounced increasing trends. However, limited consensus was also exhibited regarding the relationship between Brunt-Väisälä frequency changes and primary production changes. Therefore, we can still conclude that the local upwelling changes may overwhelm stratification changes, governing the ecosystem's productivity. Moreover, we also performed sensitivity analysis to evaluate the influence of choosing different average depths in calculating the stratification (not shown). They all show less relationship between stratification changes and primary production changes, leading supports to the validity of the conclusions in our manuscript. The above analysis has added in the revised manuscript, see lines 292-296.

Fig. R5 Same as Fig. 4c, but for the relationship between Brunt-Väisälä frequency changes and primary production changes.

Major comments:

The first question I have regarding the results in this work is related to the use of model vertical velocity outputs, as well as the use of an arbitrary depth (50 m) as the most representative of upwelling. I would like the authors to justify why they chose to use 50 m. In studies on upwelling systems, it has been agreed upon that there is significant variability both latitudinally and seasonally in the upwelling source depth. Using 50 m as a criterion oversimplifies the complexity of EBUSs. In this regard, using the MLD as the depth at each latitude and for each season would be more appropriate. However, estimating the appropriate depth remains highly complex. In fact, Jacox et al. (2018) proposes another method to assess upwelling volume that is superior to model vertical velocity outputs (Marchesiello and Estrade, 2010; Rossi et al. 2013; Oerder et al. 2015). In this context, I include the paragraph that describes this issue:

“However, we do not use modeled vertical velocities to construct the index for several reasons: First, the calculation of upwelling using modeled vertical velocity is not straightforward. One must first define a representative depth at which vertical velocity is extracted (e.g., the MLD), which will be different for each grid cell, making it difficult to close the transport budget. For example, when the MLD differs between adjacent grid cells, water can enter/exit the surface mixed layer horizontally and that component of the transport will be missed from the upwelling/downwelling estimate.”

Therefore, I would like to suggest to the authors that they reconsider the use of the MLD as the depth criterion and discuss the use of vertical velocity versus the wind field.

Marchesiello, P., and P. Estrade (2010), Upwelling limitation by onshore geostrophic flow, *J. Mar. Res.*, 68, 37–62

Rossi, V., Feng, M., Pattiaratchi, C., Roughan, M., & Waite, A. M. (2013). On the factors influencing the development of sporadic upwelling in the Leeuwin Current system. *Journal of Geophysical Research: Oceans*, 118(7), 3608-3621.

Oerder, V., Colas, F., Echevin, V., Codron, F., Tam, J., Belmadani, A., 2015. Peru-Chile upwelling dynamics under climate change. *J. Geophys. Res. Oceans* 120 (2), 1152–1172. <https://doi.org/10.1002/2014JC0102>, 99.

Jacox, M.G., Edwards, C.A., Hazen, E.L., Bograd, S.J., 2018. Coastal upwelling revisited: Ekman, Bakun, and improved upwelling indices for the U.S. West Coast. *J. Geophys. Res. Oceans* 123, 7332–7350. <https://doi.org/10.1029/2018JC014187>.

Thanks for your comments and useful references. We fully agree with you that choosing mixed layer depth (MLD) as the representative depth of vertical velocity is more appropriate for studying the upward transport of water due to the upwelling. However, we choose 50 m as the representation of upwelling in this study for the following two reasons. First, due to storage limitation, daily averaged oceanic vertical velocity is only output at selected depths (50, 105, 528 and 1146 m) in CESM-H. Considering that the MLD at the EBUSs approximates 50 m, we have therefore selected this value as the

representation depth for upwelling in our study. Second, as you pointed out, the MLD varies with latitude and seasons in the EBUSs, posing challenges in closing the transport budget due to the discontinuous distribution of MLD. Therefore, we have opted to analyze the future changes in upwelling characteristics, encompassing its timing and intensity, at a fixed depth of 50 m, which closely approximates the MLD within the EBUSs.

However, we have performed a sensitivity analysis to evaluate the influence of choosing different depths in projecting future changes in upwelling characteristics. It is worth noting that the CESM-H only provides daily vertical velocity data at a depth of 50 m, therefore, we utilized monthly data for conducting the sensitivity analysis. According to observation and our CESM-H simulation, the MLD in EBUSs ranges from 30 m to 60 m. Figure R6 displays the climatological upwelling index as well as its changes in the future at 30m, 50m, and 60m, respectively. Although some quantitative variations are observed among the upwelling changes at different depths, they consistently demonstrate an increase in UI during the month of upwelling, indicating a progressive advancement of the onset date. Similar UI changes among different depths is also detected at the termination month of upwelling across all the EBUSs. Moreover, the earlier onset and prolonged duration of upwelling season are consistently observed in CMIP6 regardless of the chosen depth (Fig. R7). The upwelling intensity changes at different depths are also compared, yielding highly consistent results that robustly support the validity of our manuscript's conclusions (Fig. R8).

Fig. R6. Climatology UI (black line) and future UI trend near transition months (orange bar) at 30 m (a-d), 50 m (e-h), and 60 m (i-l).

Fig. R7, Same as Fig. 2e-h, but for 30 m (a-d), 50 m (e-h), and 60 m (i-l).

Fig. R8 Same as Fig. 3e, but for 30 m (a), 50 m (b), and 60 m (c).

We opt for utilizing modeled vertical velocities instead of the wind-derived upwelling index due to the following reasons. First, the wind-derived upwelling index (UI_e) is depended on the assumption that coastal upwelling is majorly induced by the wind stress. This assumption could hold when the time scale is shorter, as proved in observations (Chereskin, 1995). However, over longer time scales, oceanic and atmospheric processes beyond wind stress may exert a significant influence on the upwelling variation (Ding et al., 2023; Jing et al., 2023; Rykaczewski et al., 2015). In particular, recent studies have demonstrated that oceanic processes may surpass the influence of wind stress, playing a pivotal role in modulating long-term changes in upwelling under greenhouse warming (Jing et al., 2023). Second, a key motivation of this study lies in the assessment of shifting rates of upwelling onset and termination

under greenhouse warming. However, our findings demonstrate significant disparities in estimating the upwelling onset and termination between the modeled vertical velocity and wind-derived upwelling index (Fig R9). Thus, employing the wind-derived upwelling index for assessing future changes in upwelling timing and intensity may introduce considerable uncertainties.

Fig. R9 Climatology UI derived from vertical velocity (black) and wind (red).

In the revised manuscript, we have added the sensitive analysis in selecting representative depths for assessing changes in upwelling timing and intensity. Additionally, we have emphasized the limitations of using a fixed depth instead of MLD when estimating upwelling, see line 451-458.

Chereskin, T. K. (1995). Direct evidence for an Ekman balance in the California Current.

Journal of Geophysical Research: Oceans, 100(C9), 18261–18269.

<https://doi.org/10.1029/95JC02182>

Ding, H., Alexander, M. A., & Ting, M. (2023). Revisiting the Relationship between the North Pacific High and Upwelling Winds along the West Coast of North

America in the Present and Future Climate. *Journal of Climate*, 36(23), 8211–8224. <https://doi.org/10.1175/JCLI-D-23-0238.1>

Jing, Z., Wang, S., Wu, L., Wang, H., Zhou, S., Sun, B., et al. (2023). Geostrophic flows control future changes of oceanic eastern boundary upwelling. *Nature Climate Change*, 1–7. <https://doi.org/10.1038/s41558-022-01588-y>

Rykaczewski, R. R., Dunne, J. P., Sydeman, W. J., García-Reyes, M., Black, B. A., & Bograd, S. J. (2015). Poleward displacement of coastal upwelling-favorable winds in the ocean's eastern boundary currents through the 21st century. *Geophysical Research Letters*, 42(15), 6424–6431. <https://doi.org/10.1002/2015GL064694>

The second concern is related to the domain chosen for the zonal average. In this regard, the authors average over 200 km offshore and mention that this is most appropriate for capturing both Ekman transport-associated upwelling and upwelling associated with Ekman transport divergence. Averaging over such a wide portion of the coast may obscure the upwelling transport. Perhaps with low-resolution global models, obtaining a 200 km average is necessary. However, with a model of such high resolution as proposed in this study, the authors should consider narrowing this region to a maximum of 100 km. This would allow for a more appropriate assessment of upwelling, where mesoscale processes play a significant role. Therefore, I propose that the authors conduct the analysis by limiting the average to the first 100 km of the coast. In fact, in the methodology, the authors mention the work of Du et al. (2023), who also conducted a study with an average over the first 50 km offshore. Moreover, in Fig. 1a of the manuscript, the vertical velocity associated with upwelling is closely confined to the coast.

Thanks for your comments. As you pointed out, the simulated upwelling in the four EBUSs by CESM-H consists of a rapid coastal component driven by the alongshore wind stress and confined to a narrow band (~50 km) next to the coast and a slower component driven by the wind stress curl and extending further off shore (Fig 1a), which is consistent with the existing theoretical arguments and high-resolution regional ocean simulations. The simulated upwelling by the CMIP6 models, however, exhibits

a weaker and more diffusive nature due to its coarser resolution, resulting in the absence of the prominent coastal upwelling zone. Thus, integrating vertical velocity over a sufficiently wide region should be less sensitive to model resolution and more suitable for the CMIP6 analysis. For this reason, we integrate the vertical velocity within ~ 200 km from the coast in the individual EBUSs and evaluate their long-term changes under greenhouse warming. Such integration covers the coastal upwelling and a large fraction of offshore wind stress curl-driven upwelling (Jacox et al., 2018), both of which are suggested to influence the ecosystem.

Following your advice, in the revised manuscript, we recompute the UI for the approximately 50-km-wide coastal zone in order to reveal the response of coastal upwelling to greenhouse warming. The secular changes in the timing and intensity of upwelling in the individual EBUSs resemble those calculated from the approximately 200-km-wide upwelling band from the coast (Fig. R10 and R11). The coastal upwelling within the 50-km-wide coastal zone from the coast also exhibits a seasonal advancement and prolonged duration, mirroring those found in the 200-km-wide coastal zone from the coast. Regarding the upwelling intensity, its secular changes also exhibit complicated variations across the EBUSs. However, the secular changes in individual EBUSs bear resemblance to those calculated at a distance of 200 km from the coastline, albeit with a greater magnitude. Such resemblance suggests that the response of upwelling timing and intensity under greenhouse warming is a robust feature that is not sensitive to the width of the upwelling band selected for analysis. We have added the above discussion in the revised manuscript, please see in line 128-134 and line 201-206.

Fig. R10. Same as Fig. 2a-d, but for results averaged within 50 km offshore.

Fig. R11. Same as Fig. 3a-d, but for results averaged within 50 km offshore.

Jacox, M. G., Edwards, C. A., Hazen, E. L., & Bograd, S. J. (2018). Coastal Upwelling Revisited: Ekman, Bakun, and Improved Upwelling Indices for the U.S. West Coast. *Journal of Geophysical Research: Oceans*, 123(10), 7332–7350. <https://doi.org/10.1029/2018JC014187>

Reply to the second reviewer

We are very grateful to you for your time in carefully reading our manuscript and providing helpful comments that make our manuscript better. We have carefully considered each of your comments (in blue) and revised the manuscript accordingly. Please find our response (in black) to your comments below.

Reviewer #2 (Remarks to the Author):

This paper is a model-based study (CESM-H, CMIP6, ECCO2) aimed at understanding coastal upwelling variability in the context of the climate change. While the authors have undertaken significant efforts to comprehend the seasonal progression and upwelling intensity in the EBUSs, there are concerns regarding certain processes and model biases than prevent me for approving its publication.

Major concerns:

1) Recently, it has been pointed out that the geostrophic flow may play a significant role in controlling the upwelling trends under the climate change (Jing et al., 2023). The authors should include an explanation regarding whether and why it is not being considered in this study. Could the results be the same if the geostrophic flow is considered?

Thanks for your comments. In this study, we have utilized the modelled vertical velocity (w) outputs to represent the coastal upwelling processes, which allows us to account for the contribution of geostrophic flow, as indicated in previous studies (Ding et al., 2021; Du et al., 2023; Jing et al., 2023). In accordance with your guidance, we have conducted further analyses to disentangle the impacts of wind and geostrophic flow on future upwelling changes based on CESM-H, thereby elucidating the underlying mechanisms driving future upwelling changes under greenhouse warming. It is noteworthy that the analysis presented herein pertains solely to the monthly data due to the unavailability

of daily averaged three-dimensional temperature and salinity. However, we were able to assess the upwelling timing changes by examining the variations in upwelling during the transitional month: a positive linear trend of upwelling indicates an earlier onset or a delay in termination, while a negative trend suggests a later onset or an advancement in termination. As show in Fig R1, there is a uniform increasing trend in the upwelling during upwelling onset month across all EBUSs, indicating an earlier-shifting trend of upwelling occurrence. However, the primary contributor varies across different ocean basins. In the Pacific basin, the positive trend of upwelling during the onset month is primarily attributed to changes in wind-driven Ekman transport (UI_e), illustrating the dominate role of wind variations in driving the advancement of upwelling season. In contrast, in the Atlantic basin, the geostrophic transport (UI_g) emerges as the primary contributing factor to the increasing trend in the positive trend of upwelling during the onset month. Similar situation is also detected for the upwelling changes during the terminal month of upwelling season across all the EBUSs.

Fig. R1 The climatology UI (black) and UI trend during 1920-2100 (purple) near transition months. The wind-induced UI (UI_e) trend and geostrophic-induced UI (UI_g) trend were shown in red and blue. The orange triangle demonstrates the onset of upwelling while the inverted triangle represents the termination date.

In addition, the same decomposition is also conducted for the future changes in the upwelling intensity (Fig. R2). Consistent with a recent study (Jing et al. 2023), in most EBUSs, the secular changes in upwelling intensity is primarily attributed to the geostrophic flow changes, highlighting the controlling role of geostrophic flows in upwelling intensity trends in EBUSs under greenhouse warming. However, as latitudes increase, the dominant influence of geostrophic flow gradually diminishes, yielding to the prominence of wind-induced processes. Specifically, in the poleward region of Canary Current system and Humboldt Current system, the wind-derived Ekman transport exceeds the geostrophic flow, determining the future changes in upwelling intensity.

Fig. R2. Trend of upwelling intensity calculated by UI (purple), UI_g (blue), and UI_e (red) derived from CESM-H. The error bar represents the standard error of linear regression.

We have the above analysis and discussion in the revised manuscript. See lines 156-175 and 233-243.

Ding, H., Alexander, M. A., & Jacox, M. G. (2021). Role of Geostrophic Currents in Future Changes of Coastal Upwelling in the California Current System. *Geophysical Research Letters*, 48(3), e2020GL090768. <https://doi.org/10.1029/2020GL090768>

Du, T., Wang, S., Jing, Z., Wang, H., & Wu, L. (2023). Greenhouse Warming Weakens the Seasonal Cycle of the Eastern Boundary Upwelling. *Geophysical Research Letters*, 50(11), e2023GL103857. <https://doi.org/10.1029/2023GL103857>

Jing, Z., Wang, S., Wu, L., Wang, H., Zhou, S., Sun, B., et al. (2023). Geostrophic flows control future changes of oceanic eastern boundary upwelling. *Nature Climate Change*, 1–7. <https://doi.org/10.1038/s41558-022-01588-y>

2) For the computation of the upwelling timing and intensity, the vertical velocity at 50 m depth has been considered. Could the results be different using another depth? (Note that Xiu et al. (2018) shows that, at least for CCS, there are three major drivers of the vertical velocity changes which are depth-depending: alongshore winds, wind-curl and mesoscale activity)

Following your comments, we have performed a sensitivity analysis to evaluate the influence of choosing different depths in projecting future changes in upwelling characteristics. It is worth noting that the CESM-H only provides daily vertical velocity data at a depth of 50 m; therefore, we utilized monthly data for conducting the sensitivity analysis. According to observation and our CESM-H simulation, the mixed layer depth in EBUSs ranges from 30 m to 60 m. Figure R3 displays the climatological upwelling index as well as its changes in the future at 30m, 50m, and 60m, respectively. Although some quantitative variations are observed among the upwelling changes at different depths, they consistently demonstrate an increase in UI during the month of upwelling, indicating a progressive advancement of the onset date. Similar UI changes among different depths is also detected at the termination month of upwelling across all the EBUSs. Moreover, the earlier onset and prolonged duration of upwelling season are consistently observed in CMIP6 regardless of the chosen depth (Fig. R4). The upwelling intensity changes at different depths are also compared, yielding highly consistent results that robustly support the validity of our manuscript's conclusions (Fig. R5). These has been added in line 451-458.

Fig. R3. Climatology UI (black line) and future UI trend near transition months (orange bar) at 30 m (a-d), 50 m (e-h), and 60 m (i-l).

Fig. R4, Same as Fig. 2e-h, but for 30 m (a-d), 50 m (e-h), and 60 m (i-l).

Fig. R5 Same as Fig. 3e, but for 30 m (a), 50 m (b), and 60 m (c).

3) How do the VNT and NPP appear under historical period compared to observational data (e.g., 2003-2022)? Most of the models exhibit a significant bias in the EBUSs, particularly with the biogeochemical variables. Can we have confidence in the results considering these biases?

Thanks for your comments. The sparsity of oceanic observations on ocean vertical velocity measurements leads to difficult in measure VNT directly. Therefore, we only evaluate the simulated performance of NPP by ESMS compared to the observations (Fig. R6). We agree with you that the ESMS in CMIP6 is not fine enough to well simulate the NPP. In particular, the simulated NPP in the EBUSs is weaker and more diffusive compared with the observation. Significantly, the simulation bias in NPP bears a

striking resemblance to that observed in low-resolution modes for coastal upwelling (Fig. 1 in Jing et al. 2023), suggesting that the NPP bias in ESMs may primarily stem from inadequate representation of coastal upwelling in these models. Further increasing the oceanic resolution would probably make the coastal upwelling stronger and narrower, thereby mitigating the simulation bias in NPP within the EBUSs. In addition, the low-resolution ESMs used in this study cannot resolve the oceanic small-scale processes (e.g., eddy activities, coastal trapped waves). Although resolving these small-scale processes remains probably unaccomplishable for the long-term global climate simulations up to now, they are suggested to contribute notably to the vertical nutrition transport in the mixed layer and play an important role in modulating the local NPP. Hence, the simulation bias of coastal upwelling and the deficiency in accounting for oceanic small-scale processes would both contribute to uncertainties in projecting future changes in NPP within EBUSs. We have cautioned readers about this issue in the revised manuscript. Please see Lines 303-306

Fig. R6 a, Observed vertically integrated primary production during 2003-2022, b, vertically integrated NPP derived from CMIP6 ESMs.

Minor points:

Line 111-114: Is this valid for Southern HCS and Northern HCS region? (as far as it is known, these two regions have different upwelling regimes, which one being permanent and the other seasonal)

Following your comment, we have carefully reconsidered the selection of HCS region. In the revised manuscript, the HCS region is defined from 5°S to 44°S, and further subdivided into three regions (poleward region (-P), central region (-C), and equatorward region (-E)) based on differences in upwelling seasonality, intensity as well as its potential dynamics. The HCS-E (5°S-15°S) is located at the equatorward part of the HCS, which exhibits a permanent upwelling with significant seasonal variability. Specifically, it experiences a higher magnitude during winter and a lower magnitude during summer. The HCS-C (15°S-36°S) situated at the central region of the HCS, also exhibits a strong permanent upwelling but weak seasonal variability. As for the HCS-P region (36°S-44°S), located at the poleward extent of the HCS, exhibits a seasonal upwelling phenomenon, with upwelling events predominantly occurring during summer and downwelling events observed in winter.

We have conducted a reanalysis on the future changes in the timing of the upwelling season in the subdivided regions. There is an obvious seasonal advancement and extension of coastal upwelling in the HCS-P. As for the HCS-E and HCS-C regions, the persistent annual upwelling is projected to remain throughout the 21st century, indicating no anticipated changes in the timing and duration of the upwelling season (Fig R7).

Fig. R7 The upwelling timing changes in HCS-P, HCS-C, and HCS-E

Line 235 Fig.1: why are figures 1c and 1d composed of two subfigures? (Should they be like 1f?)

This has been revised in the manuscript since we change the definition of different subregions.

Line 515 Supp. Fig.2b and 2c: the curves in these figures represent the UI only for one of the initially selected subdivisions for each upwelling system (as shown in figure 1)?

Sorry for the misleading. The UI curves shown in Supplementary Fig. 2 represent for only one subregion for each EBUSs. We have rewritten the figure caption to make it less ambiguous.

Line 547 Supp. Table 1: Include units on the table

Added.

Reply to the third reviewer

We sincerely appreciate your thorough review of our manuscript and the insightful comments you have provided. Your feedback has greatly contributed to improving the quality of our work. We have thoughtfully addressed each of your suggestions (in blue) and incorporated revisions accordingly. Below, you will find our responses (in black) to your comments.

Reviewer #3 (Remarks to the Author):

Title: Review of 'Future changes in coastal upwelling and biological production in eastern boundary upwelling systems'

The paper investigates future changes in the duration and intensity of upwelling in the eastern boundary regions and its potential consequences on primary production. Utilizing simulations from CMIP6 models and high-resolution CESM, the authors showed that the duration of upwelling is projected to be longer under high-emission scenario. But changes in the upwelling intensity show region-specific features. The authors further explore the implications of these intensity changes on net primary production. While the results are interesting and seem to be scientifically sound, this paper still needs substantial work before it can be published. I summarize my concerns and comments below.

Major concerns:

1) Refinement of introduction. This paper introduces a novel investigation into changes in upwelling duration, a distinctive aspect that warrants attention. However, it is crucial to incorporate relevant findings from existing literature to enrich the contextual background. For instance, Wang et al. (2015), utilizing CMIP5 models, observed an increased duration of upwelling in the high latitudes of major EBUSs, with differing trends in low latitudes. Summarizing these findings explicitly in the introduction is essential to provide readers with a comprehensive understanding of the advancements in this field. In addition, the introduction of future changes in net primary production

only discussed ocean stratification and upwelling intensity. However, nutrients in the source water can also impact future changes in net primary production (Rykaczewski and Dunne, 2010). Please make the introduction more comprehensive.

Thanks for your comments and useful references, we have refined the introduction to a more comprehensive version in line 42-70.

“Consequently, Bakun hypothesized that intensified land-sea thermal contrast under future warming conditions would produce stronger pressure gradients, thereby enhancing coastal upwelling in the EBUSs by fortifying alongshore winds (Bakun, 1990). However, recent studies have underscored the impact of poleward migration of the high-pressure systems on future changes in coastal upwelling, projecting an intensified upwelling in higher latitudes and weakened upwelling in lower latitudes (Rykaczewski et al., 2015). Besides, under greenhouse warming, an earlier and prolonged upwelling season was also projected in the high latitudes of EBUSs, while a different trend was identified in the low latitudes (Wang et al., 2015). These studies equate changes in upwelling with changes in the upwelling-favorable winds. However, a meta-analysis revealed substantial diversity in the responses of regional coastal upwelling to anthropogenic forcing, even under universally increasing wind trends in most EBUSs (Abrahams et al., 2021; Brady et al., 2017; Jing et al., 2023; Oyarzún & Brierley, 2019; Sousa et al., 2020; Sydeman et al., 2014; Tim et al., 2015). In particular, future changes in stratification (Oyarzún & Brierley, 2019; Sousa et al., 2020; Vázquez et al., 2023) and geostrophic transport (Ding et al., 2021; Jing et al., 2023) could surpass the impact of winds, playing an important role in modulating the coastal upwelling. Therefore, considerable uncertainty persists regarding the evolution of coastal upwelling with climate change in terms of the intensity and timing (García-Reyes et al., 2015; Wang et al., 2015).”

“It is generally thought that greenhouse warming should reduce the productivity in the EBUSs by inhibiting the upward nutrient supply due to intensified stratification, which is supported by a majority of studies (Fu et al., 2016; McGowan et al., 2003; Oyarzún & Brierley, 2019; Roemmich & McGowan, 1995; Sousa et al., 2020). However, a lack

of consensus on future productivity projections derived from climate models challenges the notion that stratification is the primary determinant in predicting forthcoming productivity changes in the EBUSs (Bograd et al., 2023). Evidence exists that the coastal upwelling contributes nonnegligible to the long-term changes of productivity in some EBUSs (Bakun et al., 2015; Xiu et al., 2018). Besides, potential alterations in the nitrate content of upwelling source water could also exert a significant influence on productivity (Renault et al., 2016; Rykaczewski & Dunne, 2010; Xiu et al., 2018). However, how the evolving dynamics of upwelling affect the nutrient supply and productivity in the major EBUSs remains shrouded in uncertainty.”

2) Refinement of Chile and Canary-Iberian upwelling systems definition. The definition of the Chile and Canary-Iberian upwelling systems requires careful consideration based on the results presented in Figure 1a. The depicted large upward vertical velocity extends beyond the designated boxes in the Chile and Canary regions, reaching substantially equatorward. Notably, the surface wind in the Chile box is perpendicular to the coast, which may not accurately represent the typical Southern HCS upwelling regions. Additionally, it is observed that stronger coast-parallel winds (as shown in Figure 1a) and higher net primary production (as indicated in Figure 1b) occur south of the defined Canary box.

In addition, if authors define upwelling regions based on vertical velocity, Figure 1a demonstrates that positive vertical velocity has very narrow zonal extents. This suggests that selecting 200 km away from the coast will include regions with negative vertical velocity. Given that upwelling timing and intensity calculations rely on box averages, the selection of these boxes has the potential to influence the results of future changes in upwelling intensity and duration. To ensure the accuracy of the study's findings, a reconsideration of the chosen boxes for defining the Chile and Canary-Iberian upwelling systems is recommended. This adjustment should aim to better capture the representative characteristics of these upwelling regions and minimize potential biases in the assessment of future changes in upwelling dynamics.

Thanks for your suggestions, we have expanded our domains according to your advice and included the definition into the methodology, please see in line 422-443.

“In terms of seasonality, each EBUS is divided into three regions: the poleward region (-P), the central region (-C), and the equatorward region (-E). The poleward regions, located at high latitudes in each EBUS, experience a pronounced seasonal upwelling, with stronger upwelling predominantly occurring during summer and weaker upwelling or downwelling observed in winter. The equatorward regions, situated at low latitudes in each EBUS, also exhibit a significant seasonal upwelling. However, the upwelling reaches its peak during summer and weakens or even shifts to downwelling during winter, in contrast to the poleward regions. The central regions, positioned between the poleward and equatorward regions, exhibit a strong permanent upwelling but display limited seasonal variability across all EBUSs.

According to the aforementioned categorization principle, the CanCS is divided into three regions: the poleward region (CanCS-P), situated along the coast of the Iberian Peninsula (36°N-43°N), the central region (CanCS-C), encompassing latitudes ranging from 19°N-36°N, and the equatorward region (CanCS-E), traditionally referred to as the Mauritania-Senegalese upwelling region, spanning from 12°N-19°N. Similarly, the HCS located in the South Pacific is also divided into three regions: HCS-E (5°S-15°S), HCS-C (15°S-36°S), and HCS-P (36°S-44°S). Likewise, the BCS situated in the South Atlantic is categorized as BCS-E (15°S-22°S), BCS-C (22°S-29°S), and BCS-P (29°S-33°S). The exception lies in the CalCS, where no winter-intensified seasonal upwelling was observed in its equatorward areas. Consequently, the CalCS region is divided solely into two regions: the central region (CalCS-C) spanning from 30°N to 40°N and the poleward region (CalCS-P) ranging from 40°N-48°N.”

For the selection of offshore distance, we initially utilized a 200-km average as necessary for the CMIP6 low-resolution models. The simulated upwelling by the CMIP6 models, exhibits a weaker and more diffusive nature due to its coarser resolution, resulting in the absence of the prominent coastal upwelling zone. Thus, integrating vertical velocity over a sufficiently wide region should be less sensitive to model

resolution and more suitable for the CMIP6 analysis. For this reason, we integrate the vertical velocity within ~ 200 km from the coast in the individual EBUSs and evaluate their long-term changes under greenhouse warming. Such integration covers the coastal upwelling and a large fraction of offshore wind stress curl-driven upwelling, both of which are suggested to influence the ecosystem.

However, in light of your feedback and to underscore the resolution advantages of CESM-H, a 200-km average may not be optimal. Therefore, in the revised manuscript, we recompute the UI for the approximately 50-km-wide coastal zone in order to reveal the response of coastal upwelling to greenhouse warming. The secular changes in the timing and intensity of upwelling in the individual EBUSs resemble those calculated from the approximately 200-km-wide upwelling band from the coast (Fig. R1 and R2). The coastal upwelling within the 50-km-wide coastal zone from the coast also exhibits a seasonal advancement and prolonged duration, mirroring those found in the 200-km-wide coastal zone from the coast. Regarding the upwelling intensity, its secular changes also exhibit complicated variations across the EBUSs. However, the secular changes in individual EBUSs bear resemblance to those calculated at a distance of 200 km from the coastline, albeit with a greater magnitude. Such resemblance suggests that the response of upwelling timing and intensity under greenhouse warming is a robust feature that is not sensitive to the width of the upwelling band selected for analysis. We have added the above discussion in the revised manuscript, please see in line 128-135 and line 201-206

Fig. R1. Same as Fig. 2a-d, but for results averaged within 50 km offshore.

Fig. R2. Same as Fig. 3a-d, but for results averaged within 50 km offshore.

3) Lack of mechanisms in upwelling changes. One notable limitation in the current study lies in the absence of a detailed exploration of the underlying mechanisms driving changes in the duration and intensity of upwelling. This omission weakens the novelty of the study, especially when compared to the work of Wang et al. (2015), who utilized CMIP5 models and already delved into discussions on these changes, demonstrating general consistency with the results presented in this paper. To strengthen the scientific contribution and novelty of this study, it is imperative to incorporate a thorough examination of the mechanisms influencing the observed changes in upwelling duration and intensity.

Appreciate for your suggestion. In accordance with your guidance, we have conducted further analyses to disentangle the impacts of wind and geostrophic flow on future upwelling changes based on CESM-H, thereby elucidating the underlying mechanisms driving future upwelling changes under greenhouse warming. It is noteworthy that the analysis presented herein pertains solely to the monthly data due to the unavailability of daily averaged three-dimensional temperature and salinity. However, we were able to assess the upwelling timing changes by examining the variations in upwelling during the transitional month: a positive linear trend of upwelling indicates an earlier onset or a delay in termination, while a negative trend suggests a later onset or an advancement in termination. As show in Fig R3, there is a uniform increasing trend in the upwelling during upwelling onset month across all EBUSs, indicating an earlier-shifting trend of upwelling occurrence. However, the primary contributor varies across different ocean basins. In the Pacific basin, the positive trend of upwelling during the onset month is primarily attributed to changes in wind-driven Ekman transport (UI_e), illustrating the

dominate role of wind variations in driving the advancement of upwelling season. In contrast, in the Atlantic basin, the geostrophic transport (UI_g) emerges as the primary contributing factor to the increasing trend in the positive trend of upwelling during the onset month. Similar situation is also detected for the upwelling changes during the terminal month of upwelling season across all the EBUSs.

Fig. R3 The climatology UI (black) and UI trend during 1920-2100 (purple) near transition months. The wind-induced UI (UI_e) trend and geostrophic-induced UI (UI_g) trend were shown in red and blue. The orange triangle demonstrates the onset of upwelling while the inverted triangle represents the termination date.

Additionally, the investigation delves into the potential dynamics of wind changes that favor upwelling in the Pacific EBUSs (Fig. R4). Following the methodology employed by Rykaczewski et al. (2015), we demonstrate a significant amplification of wind patterns during spring, which exhibits a strong correlation with variations in land-sea pressure gradients (Fig. R4 a,d). However, these pressure gradient changes are not solely induced by alterations in land-sea thermal contrast as Bakun hypothesized (Bakun, 1990). Specifically, the correlation between springtime UI_e and land-sea temperature differences did not achieve statistical significance (Fig. R4 b,e). Instead, the land-sea pressure gradients exhibit a strong correlation with the atmospheric

pressure in the interior ocean, with their inter-model correlations reaching 0.69 and 0.77 in the CalCS-P and HCS-P regions, respectively (Fig. R4 c,f). This suggests that the intensified along-shore wind during the spring season in the Pacific EBUSs might be attributed to the large-scale atmospheric circulation changes, which potentially originates from north-south shifts of the ITCZ under a warming climate (Song et al., 2018).

Fig. R4 Inter-model relationship between springtime UI_e and land-sea pressure gradients (a,d), land-sea temperature differences (b,e), and atmospheric pressure in the interior ocean (c,f) in CalCS-P and HCS-P.

In addition, the same decomposition is also conducted for the future changes in the upwelling intensity (Fig. R5). Consistent with a recent study (Jing et al. 2023), in most EBUSs, the secular changes in upwelling intensity is primarily attributed to the geostrophic flow changes. However, as latitudes increase, the dominant influence of geostrophic flow gradually diminishes, yielding to the prominence of wind-induced processes. Specifically, in the poleward region of Canary Current system and Humboldt Current system, the wind-derived Ekman transport exceeds the geostrophic flow, determining the future changes in upwelling intensity. The intensified wind stress at these high-latitude EBUSs can be attributed to the poleward migration of atmospheric systems due to greenhouse warming (Rykaczewski et al., 2015). However, the future

changes in cross-shore geostrophic flow may be influenced by various factors, including the latitudinal variation in surface warming, asymmetrical precipitation patterns, and sea ice melting in polar regions (Cheng et al., 2022; Notz & Stroeve, 2016). Consequently, it is anticipated that distinct patterns of the cross-shore geostrophic flow will manifest in individual EBUS, thereby warranting further investigation in future studies.

Fig. R5. Trend of upwelling intensity calculated by UI (purple), UI_g (blue), and UI_e (red) derived from CESM-H. The error bar represents the standard error of linear regression.

We have the above analysis and discussion in the revised manuscript. See lines 156-175 and 233-243.

Cheng, L., von Schuckmann, K., Abraham, J. P., Trenberth, K. E., Mann, M. E., Zanna, L., et al. (2022). Past and future ocean warming. *Nature Reviews Earth & Environment*, 3(11), 776–794. <https://doi.org/10.1038/s43017-022-00345-1>

Jing, Z., Wang, S., Wu, L., Wang, H., Zhou, S., Sun, B., et al. (2023). Geostrophic flows control future changes of oceanic eastern boundary upwelling. *Nature Climate Change*, 1–7. <https://doi.org/10.1038/s41558-022-01588-y>

Notz, D., & Stroeve, J. (2016). Observed Arctic sea-ice loss directly follows anthropogenic CO₂ emission. *Science*, 354(6313), 747–750. <https://doi.org/10.1126/science.aag2345>

Rykaczewski, R. R., Dunne, J. P., Sydeman, W. J., García-Reyes, M., Black, B. A., & Bograd, S. J. (2015). Poleward displacement of coastal upwelling-favorable winds in the ocean's eastern boundary currents through the 21st century. *Geophysical Research Letters*, 42(15), 6424–6431. <https://doi.org/10.1002/2015GL064694>

Song, F., Leung, L. R., Lu, J., & Dong, L. (2018). Seasonally dependent responses of subtropical highs and tropical rainfall to anthropogenic warming. *Nature Climate Change*, 8(9), 787–792. <https://doi.org/10.1038/s41558-018-0244-4>

4) Refinement of methodology concerning shifting definition in monthly data. A critical aspect requiring attention in the methodology is the method of shifting definition employed in analyzing upwelling duration based on monthly data, utilizing the epoch difference technique (Brady et al., 2017). While this method proves effective for studying the emergence of anthropogenic impacts on upwelling intensity, its applicability to upwelling duration analysis is questionable. As shown in Figure 2a-c, the onset and termination time suffer strong interannual/decadal variabilities. From 1920 to 2010, onset time varies from 01-15 to 03-01 in northern CalCS and Iberian system, and from 07-15 to 08-15 in Southern HCS system. Such variability suggests that changes in onset time using a 30-year epoch difference can be influenced by internal variability rather than anthropogenic forcing, especially when dealing with monthly data. The same problem arises from the changes in termination time. This implies that the comparison between CMIP6 and CESM-H, based on this methodology, may not be a fair one. To address this issue and ensure a more robust comparison, it is advisable to focus on confirming the role of surface wind in influencing changes in upwelling duration. Subsequently, employing daily wind data to estimate duration in CMIP6 models, as demonstrated by Wang et al. (2015), can provide a more accurate representation of the temporal dynamics of upwelling.

Thanks for your suggestion, we opt for utilizing monthly modeled vertical velocities instead of the daily wind-derived upwelling index due to the following reasons. First, a key motivation of this study lies in the assessment of shifting rates of upwelling onset and termination under greenhouse warming. However, significant disparities exist in

estimating the onset and terminational time of upwelling season between the modeled vertical velocity and wind-derived upwelling index (Fig R6), which may introduce substantial uncertainties when projecting its future changes. Second, the utilization of the wind-derived upwelling index for projecting future changes in upwelling relies on the underlying assumption that variations in coastal upwelling are predominantly driven by wind stress. This assumption could hold when the time scale is shorter, as proved in observations (Chereskin, 1995). However, over longer time scales, oceanic and atmospheric processes beyond wind stress may exert a significant influence on the upwelling variation (Ding et al., 2023; Jing et al., 2023; Rykaczewski et al., 2015). In particular, recent studies have demonstrated that oceanic processes may surpass the influence of wind stress, playing a pivotal role in modulating long-term changes in upwelling under greenhouse warming (Jing et al., 2023). Indeed, we have found that the geostrophic transport serves as the primary driver for the seasonal advancement and prolonged duration of upwelling in the Atlantic EBUSs (Fig R3). Above all, it may not be appropriate to solely consider wind changes when projecting the future changes in the timing of the upwelling season.

Fig. R6 Climatology UI derived from vertical velocity (black) and wind (red).

However, in accordance with your concerns about the influence from interannual/decadal variabilities, we extend the 30-year period to a 50-year period and redraw Fig. 2e-h (Fig. R7). The results continue to align with the that derived from the daily output in CESM-H, indicating that the observed changes could primarily represent the impact of global warming.

Fig. R7 Same as Fig. 2e-h, but for 30-year (a-d) 50-year (e-h) epoch differences.

Chereskin, T. K. (1995). Direct evidence for an Ekman balance in the California Current.

Journal of Geophysical Research: Oceans, 100(C9), 18261–18269.

<https://doi.org/10.1029/95JC02182>

Ding, H., Alexander, M. A., & Ting, M. (2023). Revisiting the Relationship between the North Pacific High and Upwelling Winds along the West Coast of North America in the Present and Future Climate. *Journal of Climate*, 36(23), 8211–8224.

<https://doi.org/10.1175/JCLI-D-23-0238.1>

Jing, Z., Wang, S., Wu, L., Wang, H., Zhou, S., Sun, B., et al. (2023). Geostrophic flows control future changes of oceanic eastern boundary upwelling. *Nature Climate Change*, 1–7.

<https://doi.org/10.1038/s41558-022-01588-y>

Rykaczewski, R. R., Dunne, J. P., Sydeman, W. J., García-Reyes, M., Black, B. A., &

Bograd, S. J. (2015). Poleward displacement of coastal upwelling-favorable winds in the ocean’s eastern boundary currents through the 21st century. *Geophysical Research Letters*, 42(15), 6424–6431.

<https://doi.org/10.1002/2015GL064694>

Minor concerns:

1) Descriptions of upwelling regions are incomplete. Firstly, the authors did not

explicitly define Northern CalCS, Iberian, and Southern HCS regions, potentially confusing readers. Secondly, the authors should describe methods to calculate the trend errors of onset and termination time that are shown in Figure 2a-c. Thirdly, the term "equatorward regions" is used without a precise definition.

Thanks for you comments, the upwelling regions' definition has been added in line 433-443.

The methods we used to calculate the linear trend and their standard errors is the Cochrane-Orcutt method (Cochrane & Orcutt, 1949), detailed description has been added in line 488-496

“The linear trends and their standard errors were obtained by regressing the time series of each upwelling metric against time using the Cochrane-Orcutt (C-O) method (Cochrane & Orcutt, 1949). This method is employed to address the autocorrelation within the time series. The model for time series of a given quantity $\theta(t)$ is expressed as $\theta(t) = \beta_0 + \beta_1 t + \epsilon(t)$, where the intercept (β_0) and linear trend (β_1) can be easily computed based on the ordinary least-squares method. Here, $\epsilon(t)$ represents a stationary stochastic process with a zero mean. The standard error of the linear trend was computed using the C-O method, assuming that the autocorrelation in $\epsilon(t)$ can be approximated as a first-order autoregressive process.”

We have reselected the subregions to poleward, central, and equatorward parts, the definition of the subdivision can be seen in line 422-431.

Cochrane, D., & Orcutt, G. H. (1949). Application of least squares regression to relationships containing auto-correlated error terms. *Journal of the American Statistical Association*, 44(245), 32–61.
<https://doi.org/10.1080/01621459.1949.10483290>

2) What are the x-axis and numbers -3, -2, ..., 2, 3 in Figure 2d-f? Authors should describe it in the caption.

Thanks for your suggestion, the figure capture has been revised more detailed.

“The positive values (+1, +2, and +3; warm color) indicate the months in which the onset and termination of the upwelling season have advanced, whereas the negative values (-1, -2, and -3; cool color) represent the months in which the onset and termination of the upwelling season have delayed. The +0 and -0 (hatched) also indicate the advancement and delay of the upwelling onset and termination but with the shifting rate less than one month. As for the duration, the positive and negative values correspond to the months that the upwelling season has prolonged and shortened, respectively.”

3) The definition of upwelling intensity in the study is based on the time and spatial integral of vertical velocity at 50 m, with onset and termination times determined using monthly data. However, an important aspect requiring clarification is the attribution of changes in upwelling intensity. The study does not explicitly specify whether alterations in upwelling intensity are primarily driven by variations in upwelling duration, vertical velocity at 50 m, or a combination of both factors.

Thanks for your suggestion. Following this comment, we have divided the upwelling intensity into two parts, the velocity-induced (the magnitude of vertical velocity at 50 m) changes and the duration-induced changes. To ascertain the impact of velocity changes on variations in upwelling intensity, we recalculated the upwelling intensity by keeping the onset date and termination date fixed at their climatological values (represented as the velocity-induced upwelling intensity). Then, the differences between the upwelling intensity and the velocity-induced intensity indicates the contribution of duration changes (duration-induced upwelling intensity).

For the annual upwelling regions, where the duration remains constant at 365 days, future changes in upwelling intensity are solely attributed to the changes in the velocity-induced upwelling intensity. In subregions where changes in duration occur, the upwelling intensity is also primarily attributed to the velocity-induced upwelling intensity changes across all EBUSs (Fig. R10). Thus, alterations in upwelling intensity are primarily driven by the changes in the magnitude of the vertical velocity.

A brief discussion has been included in the manuscript, please refer to line 232-233.

Fig. R10. Trend of changing percentage in upwelling intensity, UI-induced intensity and duration-induced intensity in CalCS-P, CanCS-P, CanCS-E, and HCS-P regions during 1920-2100. The error bar denotes the standard error of linear regression.

4) The authors consistently express the agreement percentages between CMIP6 simulations and CESM-H, but it is crucial to note that not all CMIP6 simulations were utilized in the paper. To enhance precision and clarity in reporting, it is recommended to present the results in a format that specifies the number of simulations in agreement, such as "xx out of xx simulations." This format provides a more accurate representation of the agreement statistics, ensuring that readers have a clear understanding of the scope and basis for the reported percentages.

Thanks for your helpful suggestions, this has been revised in the manuscript. Specific number of simulations were added in the brackets after the use of percentage in the manuscript.

5) CESM-H models used in this study do not include biogeochemistry components, which means that nitrate in CESM-H is a passive tracer. Given this, it is not surprising that changes in vertical velocity dominate changes in vertical nitrate fluxes, as no

additional sources and sinks of nitrate are considered in the future. To deepen the understanding of the biogeochemical dynamics, it is recommended to explore whether, in CMIP6 models with biogeochemistry components, changes in vertical nitrate flux are similarly dominated by changes in vertical velocity. This comparison will shed light on the role of biogeochemistry in influencing vertical nitrate flux patterns. In addition, total vertical nitrate flux calculated using monthly data does not include the component induced by submonthly variabilities (equivalent to eddy component). The eddy flux is resolved by CESM-H but parameterized in 1° or coarser CMIP6 models. Therefore, it is advisable to quantify the eddy fluxes, leveraging the advantages of CESM-H, to provide a more comprehensive understanding of the nuances in vertical nitrate fluxes and their implications.

Sorry about the misleading presentation. The vertical nutrient transport (VNT) and the following NPP work are all evaluated using the CMIP6 ESMs, due to the lack of biogeochemical components in CESM-H. We've added a sentence in line 249-251 for better expression.

“To test this hypothesis, we quantify the long-term changes in the upward nutrient transport and NPP based on the Earth System Models (ESMs) in CMIP6.”

We fully agree that leveraging CESM-H for quantifying eddy fluxes is indeed advantageous. However, integrating such a high-resolution model with biogeochemical components remains challenging. So, diagnosing the roles played by eddies that modulate VNT is still an ongoing work. We added some discussion about the shortcomings of the CMIP6 models to caution the readers, mainly about its spatial and temporal resolution, please see in line 303-306 and 311-314

Specific comments:

L34: EBUSs generally stands for eastern boundary upwelling systems, not coastal upwelling systems.

Revised.

L41-42: This statement is not accurate. The reason for using wind to estimate upwelling is because upwelling is mainly wind-driven, not because sparsity of oceanic observations.

Thanks for your comment. We have revised this sentence as

“Conventionally, wind-induced Ekman transport has been regarded as the primary mechanism driving upwelling in the EBUSs (Bakun, 1973).”

L47-48: Previous studies showed that projected changes in wind and upwelling both show inconsistency across different models. But in one certain model, if it is not wind determining future changes in upwelling intensity, what are the other factors? Please clarify it here.

Following your comment, the other factors has been introduced more specifically. Please see in line 53-55.

“In particular, future changes in stratification (Oyarzún & Brierley, 2019; Sousa et al., 2020; Vázquez et al., 2023) and geostrophic transport (Ding et al., 2021; Jing et al., 2023) could surpass the impact of winds, ...”

Figure 1a: vertical velocity at which depth?

Thanks for your suggestion, the vertical velocity at 50 m, this has been added in the figure caption.

L76: ‘Consistent with the existing theoretical arguments and simulations’, please add references here.

Thanks for your suggestion, the references were added.

L83: The maximum between 14°S-17°S in HCS and 27°S-28°S in BCS occurs in the early austral spring. Please precisely define poleward and equatorward regions for HCS and BCS.

Thanks for your comments, this has been revised. Please see in line 422-443 for the region definition, and the revised seasonal description can be seen in line 87-97

L86-87: Because duration of upwelling is the major focus of this paper, it will be more comprehensive to provide a figure including duration as a function of latitude in each upwelling system. It is difficult to follow the authors' argument without any figures like that.

Thanks for your suggestion. We have displayed a figure (Figure R11) imitating the Figure 1 in Wang et al. (2015) to illustrating the latitudinal distribution of duration. This has been added to the Supplementary Fig. 1.

Fig. R11. Latitudinal distribution of upwelling duration during 1920-1949. The shading demonstrating the standard error across the 30 years coverage.

L104: 'all EBUSs' means Northern CalCS, Iberian, and Southern HCS regions or regions defined by boxes in Figure 1a?

Sorry for the misleading description, this has been avoided in the revised manuscript.

L129: Authors divided EBUSs into poleward and equatorward regions, which show different changes in duration (Figure 2 and Supplementary Figure 2). 'The consistent changes across all EBUSs ...' is confusing.

Sorry for the misleading description, we have revised the word "consistent" and "all" in the manuscript.

L123-124: 75% and 73% are equivalent to how much out of how much?

Revised.

L133: 'North Atlantic'?

Sorry about the mistake, this has been revised.

L134-135: Canary and Iberian regions are not defined.

Revised, please see in line 432-443

L138: north and south portions of CalCS are not defined.

Revised, please see in line 432-443

L140-141: As shown in Figure 1, the Benguela system extends from 33°S to 15°S. Authors cannot simply group it into equatorward systems.

Thanks for your comments, we have reevaluated the subdivision of the BCS, and the description of BCS upwelling changes has been revised accordingly.

L157, 159, 161: Please add actual numbers (xx out of xx) after the percentage because you are not using all models that participated in CMIP6.

Added.

L336: Should be 'Brady et al.'

Sorry about the misspelling, this has been revised.

References:

Wang, D., Gouhier, T. C., Menge, B. A., & Ganguly, A. R. (2015). Intensification and spatial homogenization of coastal upwelling under climate change. *Nature*, 518(7539), 390-394.

Brady, R. X., Alexander, M. A., Lovenduski, N. S., & Rykaczewski, R. R. (2017). Emergent anthropogenic trends in California Current upwelling. *Geophysical Research Letters*, 44(10), 5044-5052.

Rykaczewski, R. R., & Dunne, J. P. (2010). Enhanced nutrient supply to the California Current Ecosystem with global warming and increased stratification in an earth system model. *Geophysical Research Letters*, 37(21).

REVIEWER COMMENTS

Reviewer #1 (Remarks to the Author):

The authors have done an excellent job addressing each of the questions raised in the review and implementing the various suggestions made by the reviewers. After reading the responses, I don't have any further doubts to resolve. Therefore, I propose that the article be accepted for publication in the journal.

Reviewer #3 (Remarks to the Author):

This is my second review of this paper. I appreciate the authors' efforts to enhance the introduction, refine the definition of upwelling regions, and integrate the discussions on the contributions of winds and geostrophic flow to upwelling changes. However, I cannot recommend the publication of this paper in Nature Communications with the current version. The reasons for this decision are stated as follows.

1. Some results are contradictory to previous studies. Authors show a decreasing trend in upwelling intensity in the central BCS. However, previous studies (Wang et al., 2015; Chang et al., 2023) demonstrated an increasing trend.

2. Some evidence in the paper does not support the corresponding arguments.

(1) Authors argue that results within 50 km from the coast are similar to these within 200 km. This is not entirely true. Firstly, the detected termination and onset dates are totally different, with the time discrepancy of about one month between these two cases. This discrepancy is important for policy-making in the upwelling regions. In addition, the trend of upwelling intensity in BCS-C is $-1.6 \pm 0.7\%$ per century using 50 km, but $-7.1 \pm 0.6\%$ per century using 200 km. The former one is hard to identify as a significant trend compared to the latter one.

(2) I am still not convinced that the CMIP6 results can support CESM-H results in terms of the changes upwelling durations (Fig. 2). Monthly data granularity cannot adequately capture variations of less than one month. However, using monthly data, authors identified that the changes in upwelling termination and onset dates are less than one month in some CMIP6 models. These results are questionable. Additionally, the changes in upwelling duration in CESM-H are $\sim 10.8, 21.2, 38.3, 19.3$ days per century in CalCS-P, CanCS-P, CanCS-E, and HCS-P, respectively (Fig. 2). These changes are mostly less than one month, which is impossible to be supported by monthly data.

3. The story is incomplete. The upwelling biases in CESM-H are discussed (Fig. 1) in the beginning of the paper, lending confidence to the future projections in the upwelling duration and intensity (Fig. 2 and Fig. 3). To be a complete story, the discussion of impacts of upwelling in productivity should also use CESM-H, instead of only CMIP6 models because upwelling biases in CMIP6 are not discussed in this paper. If there is no biogeochemistry component in CESM-H, I do not think it is appropriate to include the productivity results as a primary section in this work because readers will question the reliability of these findings.

4. The definitions of land-sea temperature (pressure) differences and atmospheric pressure in the interior ocean are not defined in Supplementary Figure 6. Additionally, x-axis in Figure 2e-h is still not introduced.

Reply to the third reviewer

We are very grateful to you for your time in carefully reading our revised manuscript and providing helpful comments that make our manuscript better. We have carefully considered each of your comments (in blue) and revised the manuscript accordingly. Please find our response (in black) to your comments below.

This is my second review of this paper. I appreciate the authors' efforts to enhance the introduction, refine the definition of upwelling regions, and integrate the discussions on the contributions of winds and geostrophic flow to upwelling changes. However, I cannot recommend the publication of this paper in Nature Communications with the current version. The reasons for this decision are stated as follows.

1. Some results are contradictory to previous studies. Authors show a decreasing trend in upwelling intensity in the central BCS. However, previous studies (Wang et al., 2015; Chang et al., 2023) demonstrated an increasing trend.

Thanks for your comments. As you pointed out, the upwelling intensity in the central BCS exhibits a decreasing trend, which seems to contradict previous findings by Wang et al. (2015) and Chang et al. (2023) (hereafter referred to as W15 and C23, respectively). Upon thorough investigation, we have identified the reasons for these differences in upwelling intensity trends.

The differences between our study and W15 arise primarily from variations in the data utilized to calculate upwelling intensity. W15 determined upwelling intensity using the wind stress, whereas our study used model output vertical velocity. Consequently, W15 attributed secular variations in upwelling intensity solely to Ekman transport, whereas our study considers changes from both the Ekman transport and geostrophic transport. In order to directly compare our findings with W15, we further decompose the changes in upwelling intensity into components associated with variations in wind and geostrophic flow. In particular, the wind-induced upwelling intensity (UI_e , red bar) in BCS-C in our study shows an increasing trend, consistent with the findings in W15. However, the increasing trend is counteracted by a decrease in geostrophic-induced

upwelling intensity (UI_g , blue bar), resulting in an overall negative upwelling intensity trend in BCS-C. Therefore, our findings are not contradictory to those of W15. Indeed, our study suggests that the changes in geostrophic flow may exert a significant influence on future upwelling intensity. We have added above discussion and highlight the import role of geostrophic transport on future upwelling intensity changes in the revised manuscript. See lines 241-244.

Fig. R1 Upwelling intensity percentage trends calculated by vertical velocity (purple), wind-induced upwelling index (UI_e , red) and geostrophic-induced upwelling index (UI_g , blue).

In contrast, the disparity in future upwelling intensity changes between our study and C23 primarily stem from the variation in offshore distance chosen for averaging. In particular, C23 focused on future changes in coastal upwelling within a 50-km-wide coastal zone, while we employed a 200-km average that includes both coastal upwelling and a large fraction of offshore wind stress curl-driven upwelling. Following C23, we have recomputed the upwelling intensity for the approximately 50-km-wide coastal zone in the BCS regions. The secular changes in both poleward and equatorward regions exhibit a resemblance to those calculated at a distance of 200 km from the coastline (Fig. R2a and c). However, notable differences are evident in the central

region. Specifically, we observed a negative trend in upwelling within the 200-km-wide zone across nearly all latitudes, but a distinct positive trend at specific latitudes within the 50-km-wide coastal zone, similar to the findings in Figure 3j of C23. This indicates that the secular changes in upwelling intensity exhibit a divergent pattern at varying distances from the shore in the BCS-C region.

The contour plot (Fig. R2b) provides a clearer depiction of the offshore distribution within BCS-C, showing the most significant decreases in upwelling intensity lays between 50-100 km offshore, with weaker positive trends between 20-50 km offshore. Therefore, the varying cross-shore distribution in this region is likely to be the primary factor contributing to the contrasting results between our study and C23. However, it is worth noting that the secular changes in upwelling intensity within the 50-km-wide coastal zone in most EBUSs exhibit similarities to those calculated at a distance of 200 km from the coastline. Nevertheless, we also caution readers about considering the cross-shore variations in trends in upwelling intensity in some certain EBUSs, particularly in BCS-C region. See lines 199-211.

Fig. R2 (a) Latitudinal distribution of upwelling intensity trend averaged within 200 km offshore. (b) Distribution of upwelling intensity varies with latitude and offshore distance. (c) Same as (a) but of 50-km average.

2. Some evidence in the paper does not support the corresponding arguments.

(1) Authors argue that results within 50 km from the coast are similar to these within 200 km. This is not entirely true. Firstly, the detected termination and onset dates are totally different, with the time discrepancy of about one month between these two cases. This discrepancy is important for policy-making in the upwelling regions. In addition, the trend of upwelling intensity in BCS-C is $-1.6 \pm 0.7\%$ per century using 50 km, but $-7.1 \pm 0.6\%$ per century using 200 km. The former one is hard to identify as a significant trend compared to the latter one.

Thanks for your comments. We fully agree with you that there is a discrepancy in the detected onset and termination dates between regions located within a 50-km-wide band from the coastline and those situated within a 200-km-wide band. Indeed, the climatological onset and termination date of the upwelling season changes with distance from the coast (Figure R3). However, their secular changes under greenhouse warming exhibit consistent values, thereby providing robust support for the validity of our manuscript's conclusions.

Regarding the intensity of upwelling, the long-term variations in upwelling intensity within the 50-km-wide coastal zone in most EBUSs exhibit similarities to those calculated at a distance of 200 km from the coastline, albeit with distinct magnitudes. However, we also acknowledge that the trends of upwelling intensity could exhibit a divergent pattern across cross-shore direction, particularly in BCS-C region (see our reply to your comment 1).

In this study, we seek a first-order understanding of upwelling timing and intensity change in EBUSs under global warming, which has not been previously addressed in recent studies. In addition, integrating vertical velocity over a sufficiently wide region should be less sensitive to model resolution and more suitable for the CMIP6 analysis. For these reasons, we focus on the upwelling changes averages in a 200-km-wide band from the coast. Such integration covers coastal upwelling and a large fraction of offshore wind stress curl-driven upwelling, both of which are suggested to influence the ecosystem. We know that the comprehensive understanding of the detailed structure of upwelling changes within individual EBUSs is undoubtedly important, but it exceeds

the scope of this study and warrants further investigation in future research.

We have added Figure R3 and briefly discussed the results in the revised manuscript. Please see Supplementary Figure 4, lines 130-134 and 199-211.

Fig. R3 **a-d**, The climatological and future upwelling onset and termination date in CalCS-P, CanCS-P, CanCS-E and HCS-P. The x-axis denotes the average distance from the shore.

(2) I am still not convinced that the CMIP6 results can support CESM-H results in terms of the changes upwelling durations (Fig. 2 Monthly data granularity cannot adequately capture variations of less than one month. However, using monthly data, authors identified that the changes in upwelling termination and onset dates are less than one month in some CMIP6 models. These results are questionable. Additionally, the changes in upwelling duration in CESM-H are ~10.8, 21.2, 38.3, 19.3 days per century in CalCS-P, CanCS-P, CanCS-E, and HCS-P, respectively (Fig. 2). These changes are mostly less than one month, which is impossible to be supported by monthly data.

Thanks for your comments. We admit that monthly data may not sufficiently capture precise variations in the onset and termination of upwelling, particularly for changes occurring within a time frame shorter than one month. However, due to storage limitations, daily output of vertical velocity is not provided in all CMIP6 climate models. In this study, our objective is to provide a precise characterization of the shift rates in the onset and termination of upwelling seasons using daily output from the CESM-H model. Additionally, analysis from CMIP6 serves as supplementary evidence to support the primary conclusion derived from the CESM-H model. Indeed, numerous studies have utilized monthly data to investigate the seasonal timing changes in variables such as SST (Allan & Allan, 2019; Dwyer et al., 2014; Liu et al., 2024),

precipitation (Feng et al., 2013; Song et al., 2018, 2021), and also upwelling (Brady et al., 2017; Oyarzún & Brierley, 2019; Rykaczewski et al., 2015).

In this manuscript, we assessed the changes of upwelling onset and termination by examining the variations in upwelling during the transitional month, which has been widely used in previous studies (Brady et al., 2017; Oyarzún & Brierley, 2019; Rykaczewski et al., 2015). To demonstrate that monthly data can reflect changes in upwelling timing, we conducted further analysis using three other commonly used methods:

1. Linear interpolation: linearly interpolate the monthly data to generate daily data and assess the onset and termination date of upwelling season.
2. Polynomial fitting: fit a 4th-order polynomial to the monthly data and use the polynomial to obtain the onset and termination date of upwelling season.
3. First harmonics: performing Fourier analysis on the monthly data and extracting the first harmonic mode. It is worth noting that the mean value is also important for the upwelling timing detection, therefore, not only the first harmonic mode but also the mean value (harmonic mode with zero frequency) was also included in this method to accurately determine upwelling timing.

As shown in Fig. R4, despite slight deviations observed in the reconstruction of precise upwelling onset and termination dates by each method, all approaches qualitatively reflect the secular changes derived from the model's daily output, with discrepancies less than 3 day per century for both onset and termination in most regions. We further apply these methods to the monthly output from the CMIP6 models (Fig. R5). Most CMIP6 models exhibit a seasonal advancement and extended duration of upwelling, and showing consistency across different reconstruction methods. This leads further support to our conjecture.

Fig. R4 **a-c**, Examples of the climatological results derived from direct daily output (black line), monthly output (blue dots), and reconstructed daily results via different methods (colored lines). **d-g**, Projected changes in upwelling timing derived by daily CESM-H output (black) and monthly CESM-H outputs using various methods (colors) in CalCS-P, CanCS-P, CanCS-E, and HCS-P.

Fig. R5 Projected changes in upwelling onset date (a,d,g), duration (b,e,h), and termination date (c,f,i) derived from monthly CMIP6 models in CalCS-P, CanCS-P, CanCS-E, and HCS-P. Each row represents the results from different methods.

3. The story is incomplete. The upwelling biases in CEMS-H are discussed (Fig. 1) in the beginning of the paper, lending confidence to the future projections in the upwelling duration and intensity (Fig. 2 and Fig. 3). To be a complete story, the discussion of impacts of upwelling in productivity should also use CESM-H, instead of only CMIP6 models because upwelling biases in CMIP6 are not discussed in this paper. If there is no biogeochemistry component in CEMS-H, I do not think it is appropriate to include the productivity results as a primary section in this work because readers will question the reliability of these findings.

Thanks for your suggestion. As you pointed out, the productivity in the EBUSs is heavily reliant on simulated upwelling, which may introduce significant uncertainties in low-resolution CMIP6 models due to their inherent biases in upwelling simulation. However, as highlighted by other reviewers, comprehending the response of ecosystem productivity in the EBUSs to future climate changes is a pivotal inquiry. In this study, we have unveiled an unexplored facet in the association between forthcoming alterations in upwelling and primary productivity under global warming. For this reason, we intend to retain this part in the manuscript. However, following your comments, we have reorganized the paper by moving the discussion on the implications of upwelling changes on primary productivity from the main text to the discussion section, please see in line 263-292.

Additionally, a brief discussion on the uncertainties associated with utilizing low-resolution CMIP6 data are included in line 303-311.

4. The definitions of land-sea temperature (pressure) differences and atmospheric pressure in the interior ocean are not defined in Supplementary Figure 6. Additionally, x-axis in Figure 2e-h is still not introduced.

Sorry for the negligence, we have addressed the methodological details in the figure caption of Supplementary Fig. 6. Additionally, the x-axis for Figure 2e-h has been introduced both in the figure and their captions.

REVIEWERS' COMMENTS

Reviewer #3 (Remarks to the Author):

The authors have addressed all my concerns. I will recommend the publication of this work in Nature Communications.